

# Seafloor observations at Campeche Knolls, southern Gulf of Mexico: coexistence of asphalt deposits, oil seepage, and gas venting

Heiko Sahling[1]*, Maxim Rubin Blum[2], Christian Borowski[2], Elva Escobar-Briones[3], Adriana Gaytán-Caballero[3], Chieh-Wei Hsu[1], Markus Loher[1], Ian MacDonald[4], Yann Marcon[5], Thomas Pape[1], Miriam Römer[1], Florence Schubotz[1], Daniel Smrzka[6], Gunter Wegener[1, 2], Gerhard Bohrmann[1]

[1]MARUM and Department of Geosciences at the University of Bremen, Klagenfurter Str., 28359 Bremen, Germany

[2]Max-Planck Institute for Marine Microbiology, Celsiusstr. 1, 28359 Bremen, Germany

[3]Universidad Nacional Autónoma de México, Instituto de Ciencias del Mar y Limnología, A. P. 70-305 Ciudad Universitaria, 04510 Mexico City, México

[4]Florida State University, POB 3064326, Tallahassee, FL 32306, USA

[5]Alfred Wegener Institute Helmholz Centre for Polar and Marine Research, HGF-MPG Group for Deep Sea Ecology and Technology, Am Handelshafen 12, 27570 Bremerhaven, Germany

[6]Center for Earth Sciences, University of Vienna, Althanstr. 14, 1090 Vienna, Austria

*Correspondence to*: H. Sahling (hsahling@marum.de)

**Abstract.** We studied asphalt deposits, oil seepage and gas venting during a multidisciplinary cruise in the Bay of Campeche, southern Gulf of Mexico. We conducted multibeam bathymetric mapping with an autonomous underwater vehicle and performed seafloor observations as well as sampling with a remotely operated vehicle. While previous studies concentrated on the asphalt volcano Chapopote Knoll, we confirmed that asphalt deposits at the seafloor occurred across numerous other knolls and ridges in water depths between 1230 and 3150 m; this is evidence that the outflow of heavy oil is a common component of hydrocarbon seepage of Campeche Knolls. The outflow of heavy oil either created whips or sheets floating in the water that subsequently descend and pile-up as meter high stacks at the seafloor over time or spread at the seafloor forming flows ranging from meters to tens of meters in diameter. Unlike seafloor-covering asphalts known from other continental margins, those in our study include relatively fresh material. Seafloor observations documented how chemosynthetic communities develop on the asphalts, with bacterial mats and juvenile vestimentiferan tubeworms colonizing the most recent flows.

Gas bubble emissions were an additional widespread component of hydrocarbon seepage at Campeche Knolls. The hydrocarbon gas had thermogenic origins, as indicated by the composition ($C_1/C_2$-ratio: 14 to 185) and stable carbon isotopic signature of methane ($\delta^{13}C$-$CH_4$: -45.1 to -49.8 ‰). Gas emissions were detected by multibeam echosounder at water depths as great as 3420 m over Tsanyao Yang Knoll. Gas emissions occurred at sites without large asphalt deposits (Tsanyao Yang Knoll) as well as through old, fragmented asphalts (Mictlan Knoll, Chapopote Knoll). The gas emissions feed gas hydrate deposits at shallow seafloor depth. Gas hydrate formed mounds that were ~10 m wide by several meters high in soft sediments and filled the space within fragmented asphalts. The largest gas hydrate mounds supported dense colonies of 1-2 m long



tubeworms that covered areas >100 m². These tubesworms grow with their posterior tubes implanted in a 5 to 10 cm thick reaction zone composed of authigenic carbonates, detritus, and microbial mats that overlie gas hydrate layers that were at least 2 m thick in places. This association between gas hydrates and vestimentifera has been noted in gas seeps at lesser depths, but was developed to an unequaled extent in the Campeche Knolls.

Previous studies have documented oil slicks on the ocean surface across many sites in the region. This study found liquid oil emissions in diverse settings. Sites with oil seepage are characterized by oil-soaked sediments, chemosynthetic fauna with associated heterotrophs, and bacterial coatings. Gas bubble emissions and oil seepage occurred independent of asphalt deposits or through old, fragmented asphalts, indicating that presently active hydrocarbon seepage overprints older asphalt deposits. Campeche Knolls are unique in several aspects including the occurrence of recent flows of heavy oil, deep-water hydrocarbon seepage, with many species that are new to science.

## 1 Introduction

Asphalt volcanism in the Campeche Knolls, southern Gulf of Mexico (GoM) was described as a distinct form of natural hydrocarbon seepage (MacDonald et al., 2004). Heavy oil is extruded and forms lava-like flows that cover ~100 to ~1000 m² of abyssal knolls and ridges (Brüning et al., 2010). The flows consist of high density oil with an abundant asphaltene fraction (MacDonald et al., 2004) with a terpane composition similar to what has been reported from some crude oils in the Campeche Sound (Scholz-Böttcher et al., 2008; Schubotz et al, 2011b). As described by Brüning et al. (2010), the heavy oil is at a transition point between mobile and immobile when it flows; fluid-phase material can spread smoothly over the seafloor and, because its density is initially less than seawater, local bulges and "whips" occur due to buoyancy. Once exposed to seawater, the heavy oil solidifies to brittle layers because of weathering processes and loss of volatile hydrocarbons. Fissures and cracks through the solidified asphalt deposits were observed to develop with time. Fragmentation of brittle asphalt proceeds until cobble-to-bolder sized pieces become buried by sedimentation (Brüning et al., 2010).

Most of the research conducted so far concentrated on the type locality, Chapopote Knoll, which was named after the Aztec word for tar. Because it forms a knoll with a central crater-like depression with extensive hard substrata, the term "asphalt volcano" was introduced for Chapopote Knoll (MacDonald et al., 2004). Chapopote Knoll is suggested to overlie a reservoir-seal system for hydrocarbons. From seismic studies it was inferred that coarse-grained sediments sealed by a thin (100-200 m) veneer of fine-grained sediments within the core of the knoll act as temporal hydrocarbon reservoir (Ding et al., 2008). Ding et al. (2008) speculated that the shallow reservoir plays a role in creating a heavy oil prior to its seafloor discharge: The oil in the region forms from Jurassic source material that has solidified within the hydrocarbon generation window over extended time (Magoon et al., 2001). Processes such as biodegradation, water washing or gas injection in the shallow reservoir could further increase its specific gravity (Tissot and Welte, 1984). However, Schubotz et al. (2011b) did not find evidence for biodegradation in ductile (fresh) asphalts, so the sequence leading to the characteristic asphalt deposition remains speculative. In Campeche Knolls, oil seepage is widespread as evidenced by sea surface oil slicks detected by synthetic aperture radar





satellite images (Fig.1; Williams et al., 2006). The sea surface slicks occur above dozens of knolls and ridges (Ding et al., 2010) and comprise a significant fraction of natural seepage in the southwestern Gulf of Mexico (MacDonald et al. 2015). Seismic studies at the knolls and ridges indicate shallow sub-surface hydrocarbon accumulation as well (Ding et al., 2010) but prior to the present study it was unclear whether these systems also produced asphalt deposits at the seafloor.

The natural hydrocarbon seeps at Chapopote Knoll comprise typical seep components that provide habitat for diverse biological communities. For example, in addition to asphalt deposits, there are reports of oil-soaked sediments (MacDonald et al., 2004; Schubotz et al., 2011a, b), gas hydrate occurrences (MacDonald et al., 2004, Schubotz et al., 2011b, Klapp et al., 2010a, b), gas venting (Brüning et al., 2010), and authigenic carbonates (Naehr et al., 2009). The asphalt deposits sustain hydrocarbon-degrading and sulfate-reducing bacteria within their interior (Schubotz et al., 2011b), surface bacterial mats, and vestimentiferan tubeworms that colonize fissures in the asphalt deposits (MacDonald et al., 2004; Brüning et al., 2010). Oil-impregnated sediments were found to harbor a complex microbial community that sustained both methanotrophy and methanogenesis (Schubotz et al., 2010a, b). The stable isotopes of gas hydrates associated with asphalt suggested methanogenesis (Schubotz et al., 2011b). Two species of mytilid mussel (*Bathymodiolus brooksi* and *B. haeckeri*) occurred close to an oil and gas seep, harboring different endosymbiotic communities (Raggi et al., 2013). A novel symbiont related to the hydrocarbon-degrading bacteria genus *Cycloclasticus* was verified in the seep-associated mussel *B. heckerae* (Raggi et al., 2013).

Subsequent to the initial discovery at Chapopote Knoll (MacDonald et al., 2004) asphalt deposits were found at other continental margin settings as well. In the Santa Barbara Basin off California, asphalt volcanoes about 15 and 20 m high formed 31 to 44 kyr ago (Valentine et al., 2010). These structures occur at water depths of around 150 to 180 m in a region additionally comprising tar mounds at scuba-diving depths (Vernon and Slater, 1963). In the northern Gulf of Mexico, asphalt mounds about one meter in height and up to several meters in basal diameter exist in the Puma Appraisal Area (Weiland et al., 2008) and the Shenzi Field (Williamson et al., 2008) at water depths of about 1500 m. In addition, more than 2000 asphalt mounds with diameter ranging from less than a meter up to 50 m in diameter were observed on the Angolan margin at water depth between 1350 and 2150 m (Jones et al., 2014). Based on their morphological appearance, their partial coverage by sediments, and the general lack of chemosynthetic organisms growing on the asphalt, the asphalt deposits in other regions seem to be older than those described in the southern GoM (Brüning et al., 2010). Moreover, in contrast to those in the southern GoM, asphalt deposits in the other regions are populated by regular, heterotrophic deep-sea organisms growing on hard substrate. Old asphalt deposits have, therefore, been suggested to foster the deep-sea habitat heterogeneity and provide additional settling grounds for species dispersal, which results in an increase in overall diversity (Jones et al., 2014). The Campeche Knoll asphalt region remains the deepest example known to date. The potential influence of pressure and temperature on emission of asphalt, liquid oil, and gas, and their interaction with microbial and metazoan communities, warrants further investigations. This research coincides with expanded energy exploration and production potential in the southern Gulf region.

We conducted an interdisciplinary research campaign to the Campeche Knolls in spring 2015 on board the research vessel *Meteor*. Sites for investigation were developed following a nested approach. We used the information of oil slicks at the sea



surface (Williams et al., 2006, MacDonald et al. 2015) to systematically identify potential targets with active hydrocarbon systems. To further focus our activities, we looked for evidence of gas bubble emissions by scanning the water column with ship-based multibeam echosounder. High-resolution bathymetry of the most promising sites was acquired by the SEAL 5000 autonomous underwater vehicle (AUV) mapping. Finally, we conducted seafloor observations and sampling by QUEST 4000m remotely operated vehicle (ROV), concentrating on three sites at depths > 2900m. Specifically, we were interested in disentangling the various seafloor manifestations resulting from the geochemically diverse seepage of heavy oil (leading to asphalt deposits), oil (leading to sea surface slicks) and gas (as bubbles and hydrate deposits). By concentrating on the description of the different environments that we encountered at the hydrocarbon seeps, we provide an overview for more detailed studies that will focus on the geochemistry of different oils, authigenic carbonate composition, microbiology, and fauna. We additionally show, based on our recent findings and results obtained during three precursor cruises (Bohrmann and Schenk, 2004; Bohrmann et al., 2008; MacDonald et al., 2007), that asphalt deposits are not limited to the type locality Chapopote Knoll, but are inherent to natural hydrocarbon seepage in the entire Campeche Knolls.

## 2. Geological Setting of Campeche Knolls

Campeche Knolls (Fig. 1) is part of the large South Gulf Salt Province that stretches from the southern land margin across the shelf and Campeche Knolls to Sigsbee Knolls in the north. The Sigsbee Abyssal Plain separates the salt province in the southern GoM from the Mississippi-Texas-Louisiana salt province in the north. Salt tectonism in the southern deep GoM basin is believed to be an analogue to that of the Texas–Louisiana slope (Garrison and Martin, 1973). Because the basins share similar salt deposition histories, it is inferred that most of the salt was deposited during the rifting stage of the gulf in the Late Jurassic, equivalent to the Louann salt of the Texas–Louisiana slope (Salvador, 1991).

Campeche Knolls comprise knolls and ridges that are limited to the west by smooth abyssal seafloor of the salt-free Veracruz tongue (Bryant et al., 1991). The Campeche Canyon, which is not a canyon in erosional sense, separates the eastern limit of Campeche Knolls from the carbonate Campeche Platform. Northward, smooth seafloor separates the Campeche Knolls from the Sigsbee Knolls. The Campeche Knolls is covered by a 5-10 km sediment column that hosts prolific petroleum source rocks, with the most productive being of latest Jurassic and Cretaceous age (Magoon et al., 2001). Most of the sediment thickness results from deposition during Cenozoic times and was controlled by orogenic events in the Mexican region and sea-level changes.

The distribution of oil slicks at the sea surface (Williams et al., 2006 MacDonald et al., 2015) aggregates along the northwestern sector of the Campeche Knolls (Fig. 1). Interpretation of seismic lines and basin modeling carried out by the Mexican petroleum company PEMEX led to the conclusion that the sub-province designated as 4S (Fig. 1) contains the largest allochthonous and autochthonous salt deposits (Cruz-Mercado et al., 2011; Sánchez-Rivera et al., 2011). The actual density of salt bodies increases from southeast to the northwestern region of Campeche Knolls. It is proposed that gravitational forces toward the northwest



cause contraction in the northwestern distal part of the system in which structural folds, high-angle reverse faults and numerous allochthonous salt bodies developed since the late Miocene to Recent (Cruz-Mercado et al., 2011; Sánchez-Rivera et al., 2011).

## 3 Material and Methods

Natural oil slicks at the sea surface were detected by synthetic aperture radar in the region of the southern GoM by Williams et al. (2006). Based on their analyses of repeated observations, they postulated seep locations on the seafloor that could be the origins of the oil slicks. With regard to interpretation limits of the remotely sensed data they distinguished between definite, probable and possible seep locations. Figures 1 and 2 show the inferred seep locations they classified as definite or probable. We excluded the numerous possible seep locations for clarity and also because many of the possible seeps were located outside of Campeche Knolls, suggesting that the corresponding oil slicks might not be caused by natural oil seepage. Those sea surface patterns might have been caused by other processes, i.e. algae blooms, local wind fields, or anthropogenic activity.

Ship-based hydroacoustic surveys, and dives with an autonomous underwater vehicle (AUV) and a remotely-operated vehicle (ROV) were conducted during cruise R/V *Meteor* (M 114) in February-March 2015. The multibeam echosounder (MBES) Kongsberg EM122 operating at 12 kHz was used for bathymetric mapping and water column imaging. Water column imaging for the detection of gas bubble plumes was conducted with the Fledermaus Midwater software program (company: Quality Positioning Services B. V., The Netherlands) by manually analyzing each swath. Gas bubble plumes cause high backscatter in the water column (Fig. 3). Such anomalies are commonly termed gas flares due to their appearance in echosounder records.

High-resolution bathymetry data were acquired with the MBES Kongsberg 2040 during dives with the AUV SEAL 5000 at an altitude of 80 m. The bathymetry data acquired during this cruise were combined with bathymetry data collected during two earlier cruises to the Campeche Knolls with R/V *Sonne* in 2003 (SO 174; Bohrmann and Schenk, 2004) and R/V *Meteor* (M67/2; Bohrmann et al., 2008). Ship-based and AUV-based bathymetry data were processed with MB-Systems (Caress and Cheyes, 2014). All ship-based bathymetric data were integrated into a swath map (Fig. 2). Water column data were not recorded during SO 174 and M 67/2.

The Kongsberg 675 kHz Type 1071 forward-looking sonar on ROV QUEST 4000m was used for the localization of bubble plumes following the method described in Nikolovska et al. (2008). Seafloor images shown in this study were taken either with the digital Inside Pacific Scorpio 3.3 mega-pixel still camera or frame-grabbed (Adobe Premiere Pro) from video footage gained with the Insite Pacific Zeus 3CCD HDTV video camera.

In order to characterize geochemically the different hydrocarbon components, gas bubbles (clear and oil-coated) and oil drops released from the seafloor were captured with the pressure-tight Gas Bubble Sampler at four different knolls and ridges (Supplementary Table 1). Two bubble streams were sampled at Tsanyao Yang Knoll whereas at all other sites investigated only one bubble stream was sampled. Gas bubble samples were not recovered from Knoll 2000. Repeated sampling was conducted at Mictlan Knoll during different dives and at different heights above the seafloor. Oil drops were captured together





with chimney-like structures at Marker M114-5 at a site similar in appearance to that shown in Fig. 7 D. The samples were analyzed for molecular compositions and $\delta^{13}C$-$CH_4$ signatures according to Pape et al. (2010) and Römer et al. (2012). Underwater navigation of AUV and ROV were achieved by use of the Ixsea Posidonia ultra short baseline system. Spatial data were integrated into the Arc-GIS program (ESRI) and maps were produced by combined usage of the Arc-GIS and Adobe Illustrator programs.

Camera sled surveys were conducted during cruises SO 174 (Bohrmann and Schenk, 2004), M 67/2 (Bohrmann et al., 2008), and "Chapopote III" with R/V *Justo Sierra* in 2007 (MacDonald et al., 2007). Three different camera systems were used for taking video images (M 67/2 and Chapopote III) and photographs (SO 174) at the seafloor. The locations of observations given in Table 1 are approximated based on ship position.

# 4 Results

## 4.1 Gas emissions from the seafloor

Extensive multi-beam echosounder (MBES) surveys during cruises SO 174, M 67/2, and M 114 provided a good bathymetric coverage in the entire northwestern sector of the Campeche Knolls (Fig. 2). The bathymetric map shows that knolls and ridges form separated seafloor structures in the north; whereas the morphology in the south is more complex, with coalescing ridges and domes.

We additionally examined the water column data of the MBES systematically for evidence of gas emissions along the entire cruise track of M 114 (Fig. 2, inset). We detected numerous gas bubble flares, of which some could be traced completely from the water column down to the seabed. Most flares, however, were only visualized in the mid-water column and the point of origin at the seabed could not be traced (Fig. 3). We assume that this lack of backscatter from gas bubbles in great water depths results from the pressure dependence of the backscatter strength (Greinert et al., 2006). Further in-depth studies are required to confirm this.

For the objectives of this study, we linearly projected flares from mid-water depths to the seabed and assumed that bubbles were escaping the seafloor around these sites. We defined surveys areas at the seafloor that included the putative gas emission sites and conducted AUV-based micro-bathymetric mapping and ROV-mounted horizontally scanning sonar to locate gas emission sites. This approach proved to be very efficient and resulted in the discovery of various seep manifestations at five knolls and ridges called: Tsanyao Yang Knoll, Mictlan Knoll, Chapopote Knoll, UNAM Ridge, and Knoll 2000 (locations are shown in Fig. 2).



## 4.2 Site descriptions

### 4.2.1 Tsanyao Yang Knoll (about 3400 m water depth)

Tsanyao Yang Knoll is a dome 400 m high and 6 km wide (Fig. 4 A), which we named after Prof. Tsanyao Yang from the Department of Geosciences, National University of Taiwan. "Frank" Yang dedicated his life to study mud volcanoes and hydrocarbon seeps on- and offshore Taiwan; he sadly passed away while we were conducting this present study at sea.

Flares connected to the seafloor at water depths as great as to 3500 m were recorded at the plateau-like summit (Fig. 4 A). Figure 4 B shows the ROV dive track within a depression about 150 to 200 m wide and 30 m deep. Here, we discovered spectacular gas-hydrate exposures at three sites (Site 1, 2, and 3) at depths of about 3420 m. In contrast to the knolls and ridges further south, we did not observe during the dives wide-spread asphalt deposits at Tsanyao Yang Knoll.

The gas hydrates were found in mounds that were 1-2 m high and 3-10 m wide at their bases and that were densely colonized by vestimentiferan tubeworms (Fig. 5 A). The mounds had regular borders of gravel- to bolder-sized rock clasts and shells that separated seep-influenced from normal deep-sea sediments. Occasionally, clusters of living vesicomyid clams (*Abyssogena southwardae*) were observed in the transition zone. In several instances, the mounds were dissected, exposing 1-2 m hanging walls of gas hydrate with vigorous gas bubble streams being emitted from the base (Fig. 5 B). Exposed hydrate preserved a fabric of frozen bubbles, which indicated rapid formation. Some bubble-fabric gas hydrate formed from rising bubbles when they attached to the vertical wall due to adhesion or were captured by overhanging structures. Many bubble streams were composed of transparent bubbles forming white gas hydrates but some streams contained oil-coated bubbles forming blackish-colored hydrates. We additionally observed dense hydrate outcropping over most of the exposed wall, as exemplified in Fig. 5 B. Most of the hydrate was white, but blackish-stained hydrate was also observed.

The outermost 5 to 10 cm thick layer of the mounds were composed of authigenic carbonates entangled with the posterior parts of the vestimentiferan tubes and other detritus. For convenience, we term these tubeworm parts the "rhizosphere" in the following. Fleshy, thick microbial mats were attached to the carbonates in the lower portion of the rhizosphere. We hypothesize that the carbonates were formed under anoxic conditions in the interior of the rhizosphere. A flourishing ecosystem of seep-typical fauna (e.g. living mytilid mussels, gastropods, *Alvinocaris* sp. shrimp) occupied the top of the carbonates at the base of the vestimentiferan tubes (Fig. 5 C). The vestimentiferan tubes were densely overgrown by colonies of epifaunal suspension feeders (e.g. hydrozoans and anemones).

At site 3, we observed a mound composed of reworked sediment components and gravel to pebble-sized rocks (Fig. 5 D). An exposed lens-shaped chunk of hydrate (1.5 m wide and 0.5 m high) was present at the top of the mound (Fig. 5 E). The upper part of the hydrate showed a dense fabric while the lower part was porous suggesting that it formed from frozen gas bubbles (Fig. 5 F). The two hydrate fabrics were separated by a layer that looked like organic material. Ice worms (cf. *Hesiocaeca methanicola*) were observed living in the bubble-fabric hydrate.





### 4.2.2 Mictlan Knoll (about 3100 m water depth)

Mictlan Knoll features the characteristics of an asphalt volcano (sensu MacDonald et al., 2004) with a crater-like summit area on top of a 250 m high circular knoll that is about 10 km wide at its base (Fig. 6 A). This knoll contained diverse and widespread structures related to heavy oil seepage. We named it Mictlan, which means underworld in the Aztec language.

The MBES surveys revealed evidence for gas emission along the crater rim. We conducted five ROV dives in different areas of the crater (Fig. 6 B) and illustrate our findings with images from four sites: Hydrate Hill, Marker M114-5, Marker M114-1, and "fresh asphalts".

Hydrate Hill is a 30 m high elevation with a densely populated vestimentiferan tubeworm field on its top that was 20 to 30 m wide (Fig. 7 A-C). At the summit, gas escaped the seafloor and gas hydrates formed below an overhanging protrusion consisting of carbonates and vestimentifera. Exposed rocks at the base of the hill looked like fragmented asphalts (Fig. 7 C). We therefore suggest that the entire hill is an accumulation of asphalt talus that is covered at its summit by authigenic carbonates and vestimentifera.

At Marker M114-5, we observed 1-2 m sized asphalt humps with small colonies of vestimentiferan tubeworms that were less than one meter long. Remarkably, at this site viscous oil was slowly seeping (about one drop every few minutes) through slender, white-coated chimneys some decimeters in height (Fig. 7 D). In the area around Marker M114-5 about 10 sites with chimney clusters were observed.

At Marker M114-1, oil seepage occurred through a flat-topped mound about 1 m high and tens of meters wide, which was composed of fragmented asphalt and soft sediments. The mound surface was covered by a highly patchy community of 50 cm long vestimentiferan tubeworms, mytilid bivalves (*Bathymodolus brooksi*), bacterial mats, and various epizoic groups of suspension feeders such as sponges, hydrozoa, and anemones (Fig. 7 E, F). Oil drops escaped the sediments at a low rate (a drop every few minutes). Rising drops remained at first attached to the seafloor by elongating threads and only eventually broke loose, leaving behind strands of oil that floated or adhered to the nearby organisms. Oil was released from the sediments also during sampling of sediments and organisms collected with the ROV and the sample material was heavily soaked with oil when recovered aboard ship.

In the area denoted as "fresh asphalts" (Fig. 6B) we observed a fascinating variety of structures that gave insight into the mechanism involved in heavy oil deposition at the seafloor. Although the extrusion process of heavy oil was not directly observed by us at the seafloor, the resulting flow structures are illustrative. Based on our observations, two main types of heavy-oil emission and post-emission behavior may be distinguished. The first type includes extrusion of heavy oil forming strands or sheets that float into the water due to positive buoyancy while they remain attached to the seabed owing to cohesion (Fig. 8 A-C). Over time, the strands and sheets apparently loose buoyancy and pile up on the seafloor forming decimeter to meter-high accumulations. The second type comprises extrusion of oil that is heavier than seawater and spreads on the seabed following gravity (Fig. 8 D-F). These flows solidify over time and, subsequently, accumulate sediments on their surface as



illustrated in Figs. 8 E and F. Dimensions of flows observed in this study ranged between decimeter and tens of meters in diameter.

### 4.2.3 Chapopote Knoll (about 2900 m water depth)

Flare mapping with the ship-based MBES system revealed that gas emission at this site was widespread along the crater rim (Fig. 9 A). Flares were rather dispersed; consequently, we mapped areas with gas emissions rather than individual emission sites. Subsequent ROV dives focused on the 50 m wide flow of the main asphalt field and a location that had been named the "bubble site" during previous visits (Fig. 9 B).

First analysis of results obtained during our return visit in 2015 revealed that the appearance of the main asphalt field was little altered from what was observed during the first exploration in 2006 (Brüning et al. 2010). The extent of the flow and distributions of bacterial mats and vestimentiferan tubeworm remained largely unchanged. While expanding the mapped area in the course of this study, it became apparent that the main asphalt flows terminated on soft sediments in the north and west (Fig. 10 A), but overlaid older flows characterized by fragmented asphalts (Fig. 10 B) toward the south and east. Noting significant sediment cover on older asphalts, we conjecture that the main asphalt flow represents the most recent stage of several flow events.

The "bubble site" is located on a ridge several meters high that is formed of fragmented asphalts with soft sediments interspersed. Gas bubbles escaped the seafloor and gas hydrates were present below a seafloor protrusion (Fig. 10 C, D). Recovered sediments and rocks were heavily impregnated with oil. The fauna comprised mytilid bivalves including *Bathymodiolus heckerae*, *B. brooksii*, vestimentiferan tubeworms, and a variety of organisms attracted by the seep system, such as *Munidopsis geyeri* and *Alvinocaris muricola* crustaceans (Gaytán-Caballero, 2009), gastropods and sponges (Fig. 10 D, E). A few meters away from the "bubble site", oil drops or oil-coated bubbles were released from the seafloor (Fig. 10 F) and dark orange, oil coated hydrate masses were exposed.

### 4.2.4 Knoll 2000 (about 1850 m water depth)

Knoll 2000 is an elongated feature next to a ridge with two faint flares that originated at the eastern knoll flank (Fig. 11). During an ROV dive authigenic carbonates and dark-stained sediments with whitish patches, putatively bacterial mats, were observed (Fig. 12 E, F). Some of the authigenic carbonates were dark-stained and may contained asphalt. The recovery of pogonophoran tubeworms (Siboglinidae; Anobturata) in whitish-stained sediments (Fig. 12 E) was noteworthy because species of this group were not observed at the other sites investigated in this study. However, because of their thin tubes, the tubeworms eluded detection at the seafloor during the dive. Gas bubble emissions were not observed during the one ROV dive.





### 4.2.5 UNAM Ridge (about 1230 m water depth)

UNAM ridge is a 500 m high ridge named after the *Universidad Nacional Autónoma de México* in appreciation of the collaborative effort to study the deep-water hydrocarbon seeps off Mexico. This was the shallowest site visited during this study. At least five flares were detected above the crest (Fig. 11). Two ROV dives revealed evidence for an active, albeit senescent seep system. Remarkably, soft corals and other hard-ground suspension feeders were found to settle on iron/manganese-stained carbonates and weathered asphalts (Fig. 12 A). The occurrence of vestimentiferans tubeworms was limited to a few bushes. Recumbent tubeworms were dominant (Fig. 12 B), which is generally considered as an indicator for a senescent community. Iron/manganese-stained carbonates were observed along with debris of mytild shells (Fig. 12 C). The most active seep site was situated on the side of a pockmark-like depression and comprised a relatively small bubble stream with only few specimens of living mytilid mussels on carbonates (Fig. 12 D). On a small 20 cm high and 1 m wide sediment mound we observed a white, hydrate-like texture through a drape of sediment suggesting that gas hydrates may have existed in the sediments.

### 4.3 Seafloor observations by camera sled

In addition to the findings during M 114, camera sled deployments during the preceding cruises revealed evidence for the presence of asphalts and chemosynthetic communities at seven further knolls and ridges (Tab. 1, Fig. 2). We realized that the identification of asphalts on the basis of images alone is ambiguous because the appearance of iron/manganese-stained carbonates is similar to that of weathered asphalts as shown in Figs. 12 A, C and Fig. 10 B, respectively. Nevertheless, the camera sled observations revealed that natural hydrocarbon seepage is widespread in the area of the Campeche Knolls.

### 4.4 Gas composition and isotopes

Hydrocarbon in all gas bubble samples collected with the Gas Bubble Sampler from Tsanyao Yang Knoll, Mictlan Knoll, Chapopote Knoll, and UNAM Ridge were dominated by methane with a $C_1/C_2$-ratio varying between 14 and 185 (Fig. 13). The methane stable carbon isotope composition ($\delta^{13}C$-$CH_4$) ranged between -45.1 and -49.8‰ V-PDB. The oil drop sample revealed a relative $CH_4$-depletion ($C_1/C_2 = 2$) compared to the gas samples and the $\delta^{13}C$-$CH_4$ value (-56.4‰ V-PDB) was more negative.



## 5 Discussion

### 5.1 Natural hydrocarbon seepage in the Campeche Knolls

The region of Campeche Knolls comprises abundant natural hydrocarbon seepage. This was already evident by the widespread presence of oil slicks at the sea surface (Williams et al., 2006, MacDonald et al., 2015). Knolls and ridges in Campeche Knolls are generally the seabed expressions of salt diapirism that causes hydrocarbons to accumulate and, eventually, to migrate though the sediments to the seafloor (Ding et al., 2008; 2010). More than 50 knolls and ridges in our study area exhibited single or multiple oil slick origins (Fig. 2), which suggests that it is an as prolific natural oil seepage area as the salt province in the much better studied northern Gulf of Mexico (GoM; e.g. MacDonald et al., 1993; 1996). In Campeche Knolls, hydrocarbon seepage occurs in water depths down to 3500 m, which is considerably deeper than seeps discovered at the middle-lower continental slope in the northern GoM (~2750 to ~970 m; Roberts et al., 2010). Only the seeps at the Florida Escarpment are located in comparable water depths at 3300 m (Paull et al., 1984; Cordes et al., 2007).

While prior to this study seafloor observations were only performed at Chapopote Knoll, we documented hydrocarbon seepage at the seafloor at five sites by ROV observations (including Chapopote Knoll) and, in addition, at seven sites by camera sled surveys at knolls and ridges (Fig. 2, Table 1). These sites range from depths around 1100 m in the south to 3420 m in the north of the study area. At all twelve sites we found evidence of natural hydrocarbon seepage as indicated by the presence of chemoautotrophic communities, authigenic carbonates, or asphalt deposits, confirming that hydrocarbon seepage is a widespread process in the region of Campeche Knolls. All of the twelve sites are located at the seafloor below sea surface oil slicks detected by satellite imagery (Williams et al., 2006).

Asphalt deposits were proven for seven sites (UNAM Ridge, Chapopote Knoll and Mictlan Knoll; Sites 1, 3, 6, and 7; Table 1) and they probably also occur at four more sites (Knoll 2000, Site 2, 4, and 5; Table 1). This clearly shows that seepage of heavy oil, which is a particular type of hydrocarbon seepage, is an intrinsic property of Campeche Knolls (Fig. 2). A possible reason for this is that the crude oil in the Campeche salt province is generally heavy (Magoon et al., 2001) in terms of API gravity (American Petroleum Institute gravity). For now we can only speculate on the mechanisms that lead to the expulsion and deposition of heavy oil on the seafloor, but we assume it is a combination of salt tectonic movements and high gas content that lead to the rise of this heavy oil to the seafloor, where it is then subjected to postdepositional weathering processes.

Out of the about ten sites with asphalt deposits at the seafloor, only three knolls (Chapopote Knoll, Mictlan Knoll, and Site 3; Table 1) exhibit features characteristic of "asphalt volcanism" (sensu MacDonald et al., 2004) that forms conical mounts with crater-like depressions and extensive hard substrata. The other sites either form ridges or the asphalts are associated with more complex morphologies. Tsanyao Yang Knoll is a noteworthy exception, because we did not find indications for the presence of extensive asphalt deposits, although seepage of oil was observed. The fact that we did not observe large asphalt deposits might be the result of too limited survey effort. Alternatively, we speculate that the absence of asphalt deposits at Tsanyao Yang Knoll are related to its unique shape as it is the only flat-topped knoll analyzed in this study: Its morphology resembles



that of "passive type" salt diapirs (Ding et al., 2010), where salt intruding very close to the seafloor caused bending up of hydrocarbon-bearing strata to the seabed. The structural framework of passive type salt diapirs may not provide the necessary shallow hydrocarbon reservoir typical for the other knolls and ridges (Ding et al., 2008; 2010).

During our investigations it became clear that in most instances gas venting, oil seepage, and flows of heavy oil (leading to asphalt deposits) were temporarily and spatially segregated. Oil and gas seepage occurred separated from asphalt occurrences (e.g. at Mictlan Knoll) but also through fractures in asphalts that probably act as conduits for gas/oil migration (e.g. at Hydrate Hill, Marker M114-5, Marker M114-1, and the "bubble site" at Chapopote). Different chemical compositions of oil and gas result in different manifestations at the seafloor that we discuss in the following.

**5.2 Geochemical characterization of gas, hydrate and oil**

In order to characterize the origin of hydrocarbons generated in the deep subsurface below the emission sites, we analyzed bubble-forming gas escaping the seafloor. Compared to the other hydrocarbon-containing organic substances discharged at the Campeche Knolls (e.g. oil, heavy oil), rapidly emitted vent gas is believed to be less affected by alteration in the course of upward migration in the sediment and, thus, can provide insight into the chemical characteristics of the subsurface reservoir of light hydrocarbons. Considering the molecular compositions of light hydrocarbons (expressed as $C_1/C_2$-ratios) and stable carbon isotopic signatures of methane ($\delta^{13}$C-CH$_4$) most samples plot within or close to the empirical field proposed for a thermocatalytic origin (Fig. 13).

The variability of $\delta^{13}$C-CH$_4$ values in gas bubbles was noticeably small ($\Delta\delta^{13}$C-CH$_4$ = 4.7 ‰V-PDB), given the fact that sampling was conducted at four different knolls and ridges positioned some tens to hundreds of kilometers distant to each other. In contrast, $C_1/C_2$-ratios appear to be more variable with highest values determined for apparently pure gas bubbles from Hydrate Hill at Mictlan Knoll (185, GeoB19336-15) and lowest values for oil-coated bubbles at Tsanyao Yang Knoll (14, GeoB19337-2). A variety of possible post-genetic processes can affect the distributions of light hydrocarbons, e.g., abiotic molecular fractionation during migration (Leythaeuser et al., 1980), admixture with oil-derived components or secondary methane (Milkov and Dzou, 2007), and microbial oxidation of methane and non-methane hydrocarbons (Hoehler et al., 1994; Joye et al., 2004; Kniemeyer et al., 2007). However, the causes for the variations in $C_1/C_2$-ratios observed in this study remain unexplained.

Methane in the gas bubbles sampled during this study was enriched in $^{13}$C relative to that in oil samples ($\delta^{13}$C-CH$_4$ <-50.3 ‰), which were either collected during this study (one sample) or in the course of previous studies (MacDonald et al., 2004; Schubotz et al., 2011b). The variability in the $\delta^{13}$C-CH$_4$ and $C_1/C_2$-ratios of the oil samples either reflects site-specific properties of the oil source or indicates alteration processes during migration to variable extents. More negative values $\delta^{13}$C-CH$_4$ of the oil samples in general may result from admixture by microbial produced $^{13}$C-depleted secondary methane from the oxidation of higher short-chained hydrocarbons at shallow sediment depth (Milkov and Dzou, 2007; Schubotz et al., 2011b).





We failed to sample gas hydrates during this study, but shallow hydrates were collected from two sites at Chapopote Knoll during previous investigations (MacDonald et al., 2004; Schubotz et al., 2011b). Hydrate-bound methane in these studies was depleted in $^{13}$C by > 3 ‰ compared to methane in bubbles collected in our study. It should be stressed, that sampling was conducted in different years and not at exactly the same sites. Nevertheless, $\delta^{13}$C-CH$_4$ differences between gas bubbles and hydrates within Chapopote Knoll are an interesting result, as hydrate deposits close to the seafloor are considered to form from gas bubbles (see below) without significant isotopic fractionation of methane (Bourry et al., 2009; Pape et al., 2010; Sassen et al., 2004). The difference in $\delta^{13}$C of > 3 ‰ between methane in bubble streams and in shallow hydrates at Chapopote Knoll, therefore, suggests some contribution of microbial-generated methane to the hydrate.

## 5.3 Gas venting, hydrate occurrence, and the vestimentifera-gas/hydrate habitat

Gas emissions are integral parts of submarine seep systems in various geological settings worldwide but hydrate-containing mounds overgrown by dense vestimentiferan tubeworm fields is, to our knowledge, unique to Campeche Knolls. Massive hydrate deposits close to the seafloor are considered to result from gas bubble migration through the sediments (Haeckel et al., 2004; Smith et al., 2014): Part of the gas can be sequestered as gas hydrate at shallow sediment depths, in case the crystallization force overcomes the effective overburden stress (Torres et al., 2004). Shallow hydrate deposits typically form a mounded topography of soft sediment at the seafloor as observed at Hydrate Ridge (Cascadia Margin; Suess et al., 2001; Sahling et al., 2002) and Bush Hill in the northern GoM (MacDonald et al., 1994). At Hydrate Ridge, intensive anaerobic oxidation of methane (AOM) in sediments overlying hydrates results in production of hydrogen sulfide that is consumed by sulfide-oxidizing bacteria that form mats draping the mounds (Boetius et al., 2000; Treude et al., 2003). Sulfate reduction coupled to the degradation of higher hydrocarbons brought along with oil propagating in the sediments was additionally proposed to occur in the northern GoM (Formolo et al., 2004; Joye et al., 2004).

In our study, we found vestimentiferan tubeworms growing on hydrate deposits, which has not been observed before in such a clear association. In order to illustrate the relevant processes, we sketched the two different situations encountered at Tsanyao Yang Knoll and Mictlan Knoll in Fig. 14. At both knolls, gas bubbles percolated through the mounds and we propose that continuous gas supply from below provides hydrate formation in the shallow sub-surface. At Tsanyao Yang Knoll, hydrate formed as massive layers in the sediments within the mound (Fig. 5 B) whereas for Mictlan Knoll we speculate that hydrates occupy voids between fragmented asphalt (Fig. 7 A-C). In both cases, hydrates serve as methane reservoir (e.g. Sahling et al., 2002) from which methane and other short-chained hydrocarbons are constantly diffusing towards the overlaying seawater. We further propose that AOM dominates in the rhizosphere which is a distinct 5-10 cm thick layer consisting of the posterior tubes of the vestimentiferan tubeworms. Vestimentiferan tubeworms can release sulfate through their posterior tubes (Dattagupta et al., 2006, 2008) and we therefore suggest that the rhizosphere in particular is supporting AOM. Due to the fact that AOM efficiently produce alkalinity necessary for carbonate precipitation, we propose that vestimentifera in the southern GoM largely rely on sulfide produced by sulfate-reduction coupled to methane. This is further supported by the composition



of the gas that forms the hydrate, which is dominated by methane (Fig. 13). The geochemistry in the southern GoM is in contrast to that for vestimentifera in the northern GoM, which rely on sulfide produced by sulfate-reduction coupled to methane as well as oil-derived higher hydrocarbons (e.g., Joye et al., 2004). We favor the concept of regarding the vestimentifera as ecosystem engineers (sensu Cordes et al., 2003; 2005) that play a pivotal role for this particular gas/hydrate habitat as they intensify chemical turnover processes within the rhizosphere and the space at and in between the tubes provides habitat for numerous other seep-typical species. Moreover, the thick blanket of the rhizosphere shields gas hydrate from direct contact with seawater and may impede its dissolution, thereby preserving the driver of AOM.

With regard to our discovery of the vestimentifera-gas/hydrate habitat, we attempted to characterize differences in environmental parameters in the Campeche Knolls and in areas where bacterial mats drape soft sediments covering hydrates (MacDonald et al., 1994; Suess et al., 2001). First, seeps in the southern GoM occur at greater water depth (~3000 m) than the hydrate mounds at Hydrate Ridge (~780 m) and at Bush Hill in the northern GoM (~540 m). The higher pressure at greater water depth in conjunction with very vigorous gas bubble emission, could lead to more voluminous and more rapid formation of gas hydrate close to the seafloor. In addition, the deep-water physical environment is more stable compared to that at the upper slope, where temperature and pressure changes are considerably higher (MacDonald et al., 1994, 2005). Further, with respect to the mature vestimentiferan communities and authigenic carbonates within the rhizosphere we speculate that gas seepage at our study sites was stable on timescales of hundreds of years: Vestimentifera are considered as long-living organisms, with estimated lifespans of 170-250 years for 2 m long species in the northern GoM (Bergquist et al., 2000). Although the vestimentiferan species present in the southern GoM are likely different from those in the northern GoM, it is likely that low growth is a general characteristic of vestimentifera at seeps and that the vestimentifera observed in our study (with tube length of ~2 m) are of similar age. Further support comes from a modeling study that suggested time scales of 100 to 500 years for the formation of a few cm thick authigenic carbonate crust (Luff et al., 2004).

## 5.4 Heavy oil flows leading to asphalt deposits and asphalt as habitat

Our ROV-based observations revealed numerous examples for voluminous asphalt deposits at Chapopote Knoll and Mictlan Knoll. Their formation has been explained by a model proposed by Brüning et al. (2010), which takes the API gravity into account. Oil that floats in the water while still being attached to the seabed has an API gravity slighter higher than 10°API, which corresponds to a density close to sea water. This places it at the boundary between heavy oil (10–12 to 20°API) and very heavy oil (<10–12°API), that are mobile and immobile at reservoir conditions, respectively (Tissot and Welte, 1984). Oil whips and sheets were observed (Fig. 8 A-C) indicating seepage of heavy oil, that is slightly positive buoyant (>10°API) but sufficiently cohesive to remain attached to the point of extrusion. In contrast, the extensive asphalt deposits must have been negatively buoyant (<10°API) when they exited because they clearly flowed at the seafloor (Figs. 8 D-E, Fig. 10 A). Incorporation of sediments in, and accumulation of sediments onto flowing asphalts will also contribute to their negative buoyancy. Formation of asphalt deposits, thus, depends on the viscosity of the extruded heavy to very heavy oil (~10°API) and





the duration it is exposed to weathering processes at the seafloor that lead to a transition from mobile to immobile, i.e. the viscosity is high enough to allow the ascent through the sediment, but after emission at the seafloor, it rapidly becomes immobile, probably due to the loss of volatile compounds.

In spite of this, additional possibly post-depositional processes lead to a decrease in bulk density, which can be deduced from observations at the main asphalt field at Chapopote Knoll. Pure gas hydrate, which typically exhibits a specific density less than seawater (ca. 0.9 g cm$^{-3}$), was present below and within fresh asphalts (Schubotz et al., 2011b; Klapp et al., 2010a, b) and asphalt pieces floated up when extracted from an intact flow (Brüning et al., 2010). The elevated gas content might be explained by post-depositional hydrate formation and gas invasion into the pore space of the asphalts resulting from gas supply from below. However, with respect to the considerable difference in $\delta^{13}$C signatures between hydrate methane below the asphalts (-54.8 ‰; Schubotz et al., 2011b) and the vent gas methane (-46.5 ‰; Fig. 13) it is unlikely that these gases share the same source. A possible interpretation for the more negative $\delta^{13}$C-signature of hydrate-bound methane could be that microbial methane is produced in sufficient amounts in shallow, hydrocarbon-soaked sediments below or within the asphalts.

As we did not observe active extrusion of heavy oil during the ROV dives, discharge rates of heavy oil at the seafloor (weeks to years?) remain unknown. The heavy oil apparently serves as energy source for chemosynthesis-based organisms like whitish bacterial mats and vestimentiferan tubeworms that generally depend on the supply of reduced sulfur compounds (Hilario et al., 2011; Teske & Nelson, 2006). If the outflow of heavy oil is a slow process, chemosynthetic organisms may grow while the oil is still in motion. Alternatively, the chemosynthetic organism may settle after deposition. Chemosynthetic organisms may thrive on reduced sulfur compounds contained in the heavy oil or on sulfide produced during microbial oil degradation coupled to sulfate reduction (Schubotz et al., 2011b).

Based on their distribution at the seafloor exemplified in Fig. 8 D, E, we suppose that predominantly relatively young ejections, such as whips and sheets in the water column, and those in the central parts of asphalt flows, are populated by whitish bacterial mats. Moreover, the spatial extent of bacterial mats at the main asphalt field at Chapopote Knoll during this study in 2015 was very similar to that observed during its initial documentation in 2006 (Brüning et al., 2010) demonstrating considerable ecosystem stability for many years. This is remarkable as we observed holothurians, galatheid crabs, and myriads of small crustaceans probably grazing on the bacteria. This indicates a high primary production by chemosynthetic microbes. At more distal parts of the most recent asphalt deposits, bacterial mats were absent, while tubeworms occurred nestling in fissures (Fig. 8 E and Fig. 10 A). Considering the relatively low growth rates of vestimentiferan tubeworms (Bergquist et al., 2000), we propose that even the most recent asphalt deposits discovered in this study already have existed for decades. The ostensible absence of bacteria on asphalt surface and the presence of vestimentifera growing posteriorly into the substrate suggest that the sulfide source progresses towards deeper parts of the asphalts with time. This would be in line with our concept of successive stages represented by bacteria and tubeworms, respectively.

The most common asphalt deposits in our study was in an "inert stage". It was devoid of any macro- or megafauna and ranged in terms of appearance from solid asphalt flows to partly to entirely fragmented asphalt deposits (e.g. Fig. 10 B). Most probably these asphalts do not serve as habitats for chemosynthesis-based bacteria or tubeworms anymore because a greater portion of



volatiles has already escaped to the water column. Furthermore, it is worth noticing that authigenic carbonates were virtually absent when fragmented asphalts occurred alone, i.e. without oil or gas seepage (Naehr et al., 2009). This observation suggests that microbial degradation of heavy oil does not produce sufficient alkalinity for authigenic carbonate formation. Based on our findings we may refer to the asphalt deposits without visible chemosynthetic fauna prevailing in Campeche Knolls as the "inert stage". Heterotrophic suspension feeders that are common in other regions with asphalt deposits (Weiland et al., 2008; Williamson et al., 2008; Valentine et al., 2010; Jones et al., 2014) were also absent on the inert stage at Campeche Knolls. In our study area soft corals attached to asphalts were only found at the shallowest site investigated, UNAM ridge at 1230 m water depth (Fig. 12 A). Therefore, the paucity of suspension feeders in the region of the Campeche Knolls might be caused by a limited food supply at the deep-water asphalt deposits Mictlan Knoll and Chapopote Knoll (3420 to 2900 m water depth, respectively).

In contrast to seafloor asphalt deposits in other regions (Weiland et al., 2008; Williamson et al., 2008; Valentine et al., 2010; Jones et al., 2014), those in the Campeche Knolls are sourced by geologically recent emissions of heavy oil. However, given that lobate flow patterns are still discernible for asphalts in the Santa Barbara Basin 31 to 44 kyr after seafloor deposition (Valentine et al., 2010), asphalt deposits sourced by emission of heavy oil in the Campeche Knolls may have existed for tens to probably hundred thousands of years while being fragmented and draped by sediments. The Campeche Knolls asphalts provide a natural laboratory to study their more recent genesis as well as their alteration through time.

## 5.5 Oil seepage and oil-soaked sediments as habitat

Because oil seepage is an integral component of the hydrocarbon seepage system at Campeche Knolls, as revealed by numerous oil slicks at the sea surface (Fig. 1), we concentrated on identifying oil seepage sites at the seafloor. In general, oil may rise as oil-coated gas bubbles or as oil drops (De Beukelaer et al., 2003; Leifer and MacDonald, 2003). Oily bubbles are difficult to identify visually during ascent through the water column, as they can appear as opaque as pure gas bubbles. Therefore, oil might have been a significant component of the observed gas bubble streams. Only in a few instances, the coating of gas bubbles was dark and the hydrate forming from the bubbles was dark-stained. In contrast, release of oil drops was clearly observed and occurred in association with two different seafloor manifestations: white-coated chimneys (Fig. 7 D) and oil-soaked sediments inhabited by *Bathymodiolus heckerae* (Fig. 10 F). These observations demonstrate that a seep system of oil exists next to a seep system of heavy oil leading to asphalt deposits. Seepage of oil was associated with old asphalt deposits (sediment-covered asphalt mounds and fragmented asphalt pieces) at three sites (Marker M114-1 and M115-5 at Mictlan Knoll; bubble site at Chapopote Knoll) suggesting that the oil migrated through the sediments along preexisting pathways.

At Chapopote Knoll and Mictlan Knoll biological communities were physically exposed to oil (Figs. 7 F, 10 F). As found in the northern GoM, degradation of oil-derived components in sediments, like non-methane hydrocarbons, can be performed by sulfate-reducing bacteria (belonging to the δ-proteobacteria group) while producing hydrogen sulfide (e.g. Joye et al., 2004; Kniemeyer et al., 2007). This is used as energy source for chemosynthesis-based organisms such as mat-forming, sulfide-



oxidizing bacteria and vestimentifera living in symbiosis with chemosynthetic bacteria. The two mytilid species studied in our area harbor symbionts that are capable of oxidizing sulfides as well as methane (Raggi et al., 2013). In addition, the mussel *Bathymodiolus heckerae* from the "bubble site" at Chapopote Knoll evidenced a unique symbiosis with a proteobacterium of the genus *Cycloclasticus* that is supposed to degrade hydrocarbons (Raggi et al., 2013). This symbiosis is unique to that site where the animals are virtually bathed in oil.

Preliminary interpretation of our observations at Mictlan Knoll (Fig. 7 D-F) and Chapopote Knoll (Fig. 10 D-F) suggest that the species diversity in the oil seeps is higher than at other sites characterized by gas emissions or asphalt deposits alone. We, therefore, speculate that combined gas and oil emission along with the presence of hard substrates including old fragmented asphalts provide a biogeochemical environment that promotes habitat heterogeneity and, thus, the observed species diversity.

## 5.6 Chemosynthetic communities at Campeche Knolls and future energy exploration

The natural hydrocarbon seeps of Campeche Knolls are unique in that they are located in deep waters and comprise a combination of heavy oil, oil and gas emissions. Comprehensive studies on the faunal composition are lacking so far, but our findings and previous phylogenetic results suggest a relationship of the community at Campeche Knolls and that associated to a seep system at the Florida Escarpment also situated in deep-waters (Cordes et al., 2007; Paull et al., 1984). These habitats share similar or identical species, including: Vesicomyid clam *Abyssogena southwardae* (Krylova et al., 2010), mytilid bivalves *Bathymodiolus heckerae, B. brooksi,* vestimentiferan tubeworms genetically similar to *Escarpia laminata* (Raggi et al., 2013), and crustacean decapods *Munidopsis geyeri*, *Alvinocaris muricola* (Gaytán-Caballero, 2009). The Campeche Knolls is yet another example of deep-water seeps that is important to consider in terms of biogeography of chemosynthetic communities. There is an intriguing, but up to now not well understood, connectivity between western and eastern Atlantic seep communities often referred to as the Atlantic equatorial belt province (Cordes et al., 2007). Further studies on the taxonomic composition of the deep-water Campeche Knolls fauna could shed light on questions of larval dispersal and connectivity with other deep-water seeps such as those off Barbados (Olu et al., 1996) and western Africa (Olu et al., 2010).

The ecosystem in the Campeche Knolls is so far only marginally affected by human activity, including our own scientific research. We limited our impact by only retrieving small amounts of samples preferentially taken by ROV manipulator. During our ROV seafloor operations, we observed anthropogenic litter (mainly plastic bags) on the seafloor in about ten instances (Fig. 15). Although we believe that these do not pose severe threats on the seep communities, the litter shows that there is hardly any untouched ecosystem left on earth.

Future exploration for oil, however, can have a major impact on chemosynthetic ecosystems through mechanical disturbance by anchors to hold drilling rigs in position and pipelines. With the oil exploration shifting further off-shore to greater water depths, the unique ecosystems may be in danger in the future. We, therefore, call for protective policies when it comes to exploration activities at or near chemosynthesis-based ecosystems in Campeche Knolls. Energy exploration and production actively avoids sites containing shallow gas and similar hazards, so protection of Campeche Knoll sites would be consistent





with existing industry practices. Moreover, drilling operations that encounter asphalts at the seafloor or in the subbottom will face significant challenges. So future expansion of the Mexican ultra-deep energy industry will require ongoing scientific research.

# 6 Conclusion

Natural oil seepage is inherent to Campeche Knolls, as revealed by our new findings obtained during cruise M114 in 2015, and by a reanalyzes of seafloor images gathered during previous cruises. Unique to Campeche Knolls is the widespread occurrence of asphalt deposits, which was definitely confirmed at seven sites. The flow structures of heavy oil encountered at Chapopote Knoll and Mictlan Knoll are noteworthy, as they represent more recent asphalt deposits when compared to those described from other continental margin settings. The recently discharged asphalts sustain chemosynthetic organisms such as bacterial mats and vestimentifera as well as a suite of heterotrophic organisms that warrant further taxonomic studies. Based on our preliminary observations, and in contrast to the asphalt deposits at other continental margins, those at the deeper Campeche Knolls (>2900 m water depth) are generally not utilized by sessile, filter-feeding organisms that attach to hard substrates. This might be caused by limited food availability at the deep-sea sites investigated.

Seepage of oil and gas bubbles co-occurs next to the asphalt deposits. Most intriguing was our finding of vigorous gas bubble streams forming hydrate mounds (Tsanyao Yang Knoll) or percolating through old, fragmented asphalts leading to hydrate precipitation in the voids between the asphalt breccia (Mictlan Knoll). The hydrates likely serve as shallow methane reservoir underlying dense communities of vestimentiferan tubeworms, which then act as ecosystem engineers, facilitating AOM and preserving the hydrate. In the tubeworms rhizosphere intense microbially-mediated turn-over processes are probably taking place that cause precipitation of authigenic carbonates. Our observation of the closely associated hydrate and vestimentiferan tubeworm is unparalleled, but this relationship and the involved biogeochemical processes could also be relevant in other hydrocarbon settings with shallow hydrate deposits. The healthy-appearing vestimentifera growing on hydrate suggest considerable stability of these habitats over time spans of hundreds of years, required for establishment of the slow-growing vestimentifera and formation of authigenic carbonates.

Oil drops escaped the seafloor through small chimney-like structures, or, together with gas bubbles, through a mixture of fragmented, old asphalt and sediments. In the latter case, sediments were heavily impregnated by oil. The seep-associated communities appeared very diverse, with two chemosynthetic mussel species and various other heterotrophic organisms. We call for protective policies to avoid negative impacts by future oil exploration on the natural hydrocarbons seeps, which provide habitats for unique deep-water ecosystems in the Campeche Knolls.



**Author contribution**

Maxim Rubin Blum, Gerhard Bohrmann, Christian Borowski, Chieh-Wei Hsu, Markus Loher, Ian MacDonald, Yann Marcon, Miriam Römer, Heiko Sahling, Florence Schubotz, Daniel Smrzka, and Gunter Wegener conducted the ROV dives as scientific advisors on which this present study is largely based on. Chieh-Wei Hsu, Markus Loher, and Miriam Römer detected the gas emissions by hydroacoustic means. Thomas Pape analyzed the gas. Adriana Gaytán-Caballero assisted in taxonomic determination of fauna at sea. Elva Escobar-Briones supported the application procedure to acquire the Mexican research permission and supervised the participation of three Mexican scientists and students in the cruise. Heiko Sahling prepared the manuscript with contributions of all co-authors.

**Acknowledgement**

We are grateful to the Mexican authorities for granting permission to conduct the research in the southern Gulf of Mexico (permission of DGOPA: 02540/14 from 5 November 2014). We thank the master and crew of R/V *Meteor* for their highly professional assistance at sea. Thanks to MARUM AUV SEAL 5000 and ROV QUEST 4000m teams for providing and handling of the indispensable equipment at sea. Various people on board have contributed to the successful cruise, we namely thank Monika Breitzke, Stefanie Buchheister, Christian Ferreira, Patrizia Geprägs, Jan-Derk Groeneveld, Elvira Jiménez Guadarrama, Ingo Klaucke, Sven Klüber, Esmeralda Morales Dominguez, Marta Torres, Monika Wiebe, Paul Wintersteller, and Jennifer Zwicker. The cruise was core funded by the German Research Foundation (DFG – Deutsche Forschungsgemeinschaft) through the cruise proposal "Hydrocarbons in the southern Gulf of Mexico". Additional support was provided through the DFG-Research Center / Excellence Cluster "The Ocean in the Earth System".

**Literature**

Bergquist, D. C., Williams, F. M., and Fisher, C. R.: Longevity record for deep-sea invertebrate, Nature, 403, 499-500, 2000.

Boetius, A., Ravenschlag, K., Schubert, C. J., Rickert, D., Widdel, F., Gieseke, A., Amann, R., Jørgensen, B. B., Witte, U., and Pfannkuche, O.: A marine microbial consortium apparently mediating anaerobic oxidation of methane, Nature, 407, 623-626, 2000.

Bohrmann, G. and Schenk, S.: RV *Sonne*. Cruise report SO 174, OTEGA II: (Lotus-Omega-Mumm). Balboa-Corpus Christi-Miami. October 1 - November 12, 2003, 117 pp., 2004.

Bohrmann, G., Spiess, V., and cruise participants: Report and preliminary results of R/V *Meteor* Cruise M67/2a and 2b, Balboa - Tampico - Bridgetown, 15 March - 24 April, 2006. Fluid seepage in the Gulf of Mexico., Berichte Fachbereich Geowissenschaften, Universität Bremen, Bremen, 263, 119 pp., 2008.



Bourry, C., Chazallon, B., Charlou, J. L., Donval, J. P., Ruffine, L., Henry, P., Geli, L., and Cagatay, M. N.: Free gas and gas hydrates from the Sea of Marmara, Turkey. Chemical and strucural characterization, Chemical Geology, 264, 192-206, 2009.

Brüning, M., Sahling, H., MacDonald, I. R., Ding, F., and Bohrmann, G.: Origin, distribution, and alteration of asphalts at Chapopote Knoll, Southern Gulf of Mexico, Marine and Petroleum Geology, 27, 1093-1106, 2010.

Bryant, R. B., Lugo, J., Córdova, C., and Salvador, A.: Physiography and bathymetry. In: The Geology of Northern America, The Gulf of Mexico Basin, Salvador, A. (Ed.), Geological Society of America, Boulder, Colorado, 1991.

Caress, D. W. and Cheyes, D. N.: http://www.mbari.org/data/mbsystem/, last access: 31 January 2014, 2014.

Cordes, E. E., Bergquist, D. C., Shea, K., and Fisher, C. R.: Hydrogen sulphide demand of long-lived vestimentiferan tube worm aggregations modifies the chemical environment at deep-sea hydrocarbon seeps, Ecology Letters, 6, 212-219, 2003.

Cordes, E. E., Carney, S. L., Hourdez, S., Carney, R. S., Brooks, J. M., and Fisher, C. R.: Cold seeps of the deep Gulf of Mexico: Community structure and biogeographic comparison to Atlantic equatorial belt seep communities, Deep-Sea Research I, 54, 637-653, 2007.

Cordes, E. E., Hourdez, S., Predmore, B. L., Redding, M. L., and Fisher, C. R.: Succession of hydrocarbon seep communities associated with the long-lived foundation species *Lamellibrachia luymesi*, Marine Ecology Progress Series, 305, 17-29, 2005.

Cruz-Mercado, M. Á., Flores-Zamora, J. C., León-Ramirez, R., López-Céspedes, H. G., Peterson-Rodríguez, R. H., Reyes-Tovar, E., Sánchez-Rivera, R. S., and Barrera-Gonzáles, D.: Salt provinces in the Mexican portion of the Gulf of Mexico - structural characterization and evolutionary model, Gulf Coast Association of Geological Societies Transactions, 61, 93-103, 2011.

Dattagupta, S., Arthur, M. A., and Fisher, C. R.: Modification of sediment geochemistry by the hydrocarbon seep tubeworm *Lamellibrachia luymesi*: A combined empirical and modeling approach, Geochimica et Cosmochimica Acta, 72, 2298-2315, 2008.

Dattagupta, S., Miles, L. L., Barnabei, M. S., and Fisher, C. R.: The hydrocarbon seep tubeworm *Lamellibrachia luymesi* primarily eliminates sulfate and hydrogen ions across its roots to conserve energy and ensure sulfide supply, The Journal of Experimental Biology, 209, 3795-3805, 2006.

De Beukelaer, S. M., MacDonald, I. R., Guinnasso, N. L., and Murray, J. A.: Distinct side-scan sonar, RADARSAT SAR, and acoustic profiler signatures of gas and oil seeps on the Gulf of Mexico slope, Geo-Marine Letters, 23, 177-186, 2003.

Ding, F., Spiess, V., Brüning, M., Fekete, N., Keil, H., and Bohrmann, G.: A conceptual model for hydrocarbon accumulation and seepage processors around Chapopote asphalt site, southern Gulf of Mexico: From high resolution seismic point of view, Journal of Geophysical Research, 113, B08404, 2008.

Ding, F., Spiess, V., MacDonald, I. R., Brüning, M., Fekete, N., and Bohrmann, G.: Shallow sediment deformation styles in north-west Campeche Knolls, Gulf of Mexico and their controls on the occurrence of hydrocarbon seepage, Marine and Petroleum Geology, 27, 959-972, 2010.



Formolo, M. J., Lyons, T. W., Zhang, C., Kelley, C., Sassen, R., Horita, J., and Cole, D. R.: Quantifying carbon sources in the formation of authigenic carbonates at gas hydrate sites in the Gulf of Mexico, Chemical Geology, 205, 253-264, 2004.

Garrison, L. E. and Martin, R. G. J.: Geologic structure in the Gulf of Mexico, U. S. Government Printing Office, Washington, D. C., 1-85 pp., 1973.

Gaytán-Caballero, A.: *Munidopsis geyeri* Pequegnat & Pequegnat, 1970 asociado al volcán de asfalto (sur del Golfo de México) y su vinculación con las poblaciones del Atlántico, master thesis, Universidad Nacional Autónoma de México, Mexico City, 146pp, 2009.

Greinert, J., Artemov, Y., Egorov, V., Debatist, M., and McGinnis, D.: 1300-m-high rising bubbles from mud volcanoes at 2080m in the Black Sea: Hydroacoustic characteristics and temporal variability, Earth and Planetary Science Letters, 244, 1-15, 2006.

Haeckel, M., Suess, E., Wallmann, K., and Rickert, D.: Rising methane gas bubbles form massive hydrate layers at the seafloor, Geochimica et Cosmochimica Acta, 68, 4335-4345, 2004.

Hilario, A., Capa, M., Dahlgren, T. G., Halanych, K. M., Little, C. T., Thornhill, D. J., Verna, C., and Glover, A. G.: New perspectives on the ecology and evolution of siboglinid tubeworms, PLoS One, 6, e16309, 2011.

Hoehler, T. M., Alperin, M. J., Albert, D. B., and Martens, C. S.: Field and laboratory studies of methane oxidation in an anoxic marine sediment: evidence for methanogenic-sulfate reducer consortium, Global Biogeochemical Cycles, 8, 451-463, 1994.

Jones, D. O. B., Walls, A., Clare, M., Fiske, M. S., Weiland, R. J., O'Brien, R., and Touzel, D. F.: Asphalt mounds and associated biota on the Angolan margin, Deep Sea Research Part I: Oceanographic Research Papers, 94, 124-136, 2014.

Joye, S. B., Boetius, A., Orcutt, B. N., Montoya, J. P., Schulz, H. N., Erickson, M. J., and Lugo, S. K.: The anaerobic oxidation of methane and sulfate reduction in sediments from Gulf of Mexico cold seeps, Chemical Geology, 205, 219-238, 2004.

Klapp, S. A., Bohrmann, G., Kuhs, W. F., Mangir Murshed, M., Pape, T., Klein, H., Techmer, K. S., Heeschen, K. U., and Abegg, F.: Microstructures of structure I and II gas hydrates from the Gulf of Mexico, Marine and Petroleum Geology, 27, 116-125, 2010a.

Klapp, S. A., Murshed, M. M., Pape, T., Klein, H., Bohrmann, G., Brewer, P. G., and Kuhs, W. F.: Mixed gas hydrate structures at the Chapopote Knoll, southern Gulf of Mexico, Earth and Planetary Science Letters, 299, 207-217, 2010b.

Kniemeyer, O., Musat, F., Sievert, S. M., Knittel, K., Wilkes, H., Blumenberg, M., Michaelis, W., Classen, A., Bolm, C., Joye, S. B., and Widdel, F.: Anaerobic oxidation of short-chain hydrocarbons by marine sulphate-reducing bacteria, Nature, 449, 898-902, 2007.

Krylova, E. M., Sahling, H., and Janssen, R.: *Abyssogena*: A new genus of the family Vesicomyidae (Bivalvia) from deep-water vents and seeps, Journal of Molluscan Studies, 76, 107-132, 2010.

Leifer, I. and MacDonald, I.: Dynamics of the gas flux from shallow gas hydrate deposits: interaction between oily hydrate bubbles and the oceanic environment, Earth and Planetary Science Letters, 210, 411-424, 2003.

Leythaeuser, D., Schaefer, R.G., and Yukler, A.: Diffusion of light hydrocarbons through near-surface rocks, Nature, 284, 522-525, 1980.





Luff, R., Wallmann, K., and Aloisi, G.: Numerical modeling of carbonate crust formation at cold vent sites: Significance for fluid and methane budgets and chemosynthetic biological communities, Earth and Planetary Science Letters, 221, 337-353, 2004.

MacDonald, I., Bender, L., Vardaro, M., Bernard, B., and Brooks, J.: Thermal and visual time-series at a seafloor gas hydrate deposit on the Gulf of Mexico slope, Earth and Planetary Science Letters, 233, 45-59, 2005.

MacDonald, I. R., Bohrmann, G., Escobar, E., Abegg, F., Blanchon, P., Blinova, V., Brückmann, W., Drews, M., Eisenhauer, A., Han, X., Heeschen, K., Meier, F., Mortera, C., Naehr, T., Orcutt, B., Bernard, B., Brooks, J., and de Farágo, M.: Asphalt volcanism and chemosynthetic life in the Campeche Knolls, Gulf of Mexico, Science, 304, 999-1002, 2004.

MacDonald, I. R., Escobar, E., Naehr, T., Joye, S., and Spiess, V.: The asphalt ecosystem of the Gulf of Mexico: Results from the Chapopte III Cruise, Fall Meet. Suppl., Abstract B43E-1660, 2007.

MacDonald, I. R., Garcia-Pineda, O., Beet, A., Daneshgar Asl, S., Feng, L., Graettinger, G., French-McCay, D., Holmes, J., Hu, C., Huffer, F., Leifer, I., Muller-Karger, F., Solow, A., Silva, M., and Swayze, G.: Natural and unnatural oil slicks in the Gulf of Mexico, Journal of Geophysical Research: Oceans, 120, 8364-8380, 2015.

MacDonald, I. R., Guinasso jr., N. L., Ackleson, S. G., Amos, J. F., R., D., Sassen, R., and Brooks, J. M.: Natural oil slicks in the Gulf of Mexico visible from space, Journal of Geophysical Research, 98, 16351-16364, 1993.

MacDonald, I. R., N. L. Guinasso, J., Sassen, R., Brooks, J. M., Lee, L., and Scott, K. T.: Gas hydrate that breaches the sea floor on the continental slope of the Gulf of Mexico, Geology, 22, 699-702, 1994.

MacDonald, I. R., Reilly, J. F., Jr., Best, S. E., Venkataramaiah, R., Sassen, R., and Guinasso, N. L., Jr.: Remote sensing inventory of active oil seeps and chemosynthetic communities in the northern Gulf of Mexico. In: Hydrocarbon migration and its near-surface expression, Schumacher, D. and Abrams, M. A. (Eds.), American Association of Petroleum Geologists, 27-37, 1996.

Magoon, L. B., Hudson, T. L., and Cook, H. E.: Pimienta-Tamabra (!) - A giant supercharged petroleum system in the southern Gulf of Mexico, onshore and offshore Mexico. In: The western Gulf of Mexico basin: Tectonics, sedimentary basins, and petroleum systems, Bartolini, C., Biuffler, R. T., and Cantú-Chapa, A. (Eds.), AAPG Memoir, 75, 83-125, 2001.

Milkov, A.V. and Dzou, L,: Geochemical evidence of secondary microbial methane from very slight biodegradation of undersaturated oils in a deep hot reservoir, Geology, 35, 455-458, 2007.

Naehr, T. H., Birgel, D., Bohrmann, G., MacDonald, I. R., and Kasten, S.: Biogeochemical controls on authigenic carbonate formation at the Chapopote "asphalt volcano", Bay of Campeche, Chemical Geology, 266, 390-402, 2009.

Nikolovska, A., Sahling, H., and Bohrmann, G.: Hydroacoustic methodology for detection, localization, and quantification of gas bubbles rising from the seafloor at gas seeps from the Black Sea, Geochemistry Geophysics Geosystems, 9, Q10010, 2008.

Olu, K., Cordes, E. E., Fisher, C. R., Brooks, J. M., Sibuet, M., and Desbruyeres, D.: Biogeography and potential exchanges among the atlantic Equatorial belt cold-seep faunas, PLoS One, 5, e11967, 2010.




Olu, K., Sibuet, M., Harmegnies, F., Foucher, J.-P., and Fiala-Médioni, A.: Spatial distribution of diverse cold seep communities living on various diapiric structures of the southern Barbados prism, Progress in Oceanography, 38, 347-376, 1996.

Pape, T., Bahr, A., Rethemeyer, J., Kessler, J. D., Sahling, H., Hinrichs, K.-U., Klapp, S. A., Reeburgh, W. S., and Bohrmann, G.: Molecular and isotopic partitioning of low-molecular-weight hydrocarbons during migration and gas hydrate precipitation in deposits of a high-flux seepage site, Chemical Geology, 269, 350-363, 2010.

Paull, C. K., Hecker, B., Commeau, R., Freeman-Lynde, R. P., Neumann, C., Corso, W. P., Golubic, S., Hook, J. E., Sikes, E., and Curray, J.: Biological communities at the Florida Escarpment resemble hydrothermal vent taxa, Science, 226, 965-967, 1984.

Raggi, L., Schubotz, F., Hinrichs, K. U., Dubilier, N., and Petersen, J. M.: Bacterial symbionts of *Bathymodiolus* mussels and *Escarpia* tubeworms from Chapopote, an asphalt seep in the southern Gulf of Mexico, Environmental Microbiology, 15, 1969-1987, 2013.

Roberts, H. H., Shedd, W., and Hunt Jr, J.: Dive site geology: DSV ALVIN (2006) and ROV JASON II (2007) dives to the middle-lower continental slope, northern Gulf of Mexico, Deep Sea Research Part II: Topical Studies in Oceanography, 57, 1837-1858, 2010.

Römer, M., Sahling, H., Pape, T., Bahr, A., Feseker, T., Wintersteller, P., and Bohrmann, G.: Geological control and magnitude of methane ebullition from a high-flux seep area in the Black Sea-the Kerch seep area, Marine Geology, 319-322, 57-74, 2012.

Sahling, H., Rickert, D., Lee, R. W., Linke, P., and Suess, E.: Macrofaunal community structure and sulfide flux at gas hydrate deposits from the Cascadia convergent margin, NE Pacific, Marine Ecology Progress Series, 231, 121-138, 2002.

Salvador, A. (Ed.): The Gulf of Mexico basin, Geology of North America, Geological Society of America, Boulder, Colorado, 568p, 1991.

Sánchez-Rivera, R. S., Cruz-Mercado, M. Á., Reyes-Tovar, E., López-Céspedes, H. G., Peterson-Rodriguez, R. H., Flores-Zamora, J. C., Ramirez, R. L., and Barrera-Gonzáles, D.: Tectonic evolution of the South Gulf Salt Province in the Gulf of Mexico, Gulf Coast Association of Geological Societies Transactions, 61, 421-427, 2011.

Sassen, R., Roberts, H. H., Carney, R., Milkov, A. V., DeFreitas, D. A., Lanoil, B., and Zhang, C.: Free hydrocarbon gas, gas hydrate, and authigenic minerals in chemosynthetic communities of the northern Gulf of Mexico continental slope: relation to microbial processes, Chemical Geology, 205, 195-217, 2004.

Scholz-Böttcher, B. M., Ahlf, S., Vazquez-Gutierrez, F., and Rullkötter, J.: Sources of hydrocarbon pollution in surface sediments of the Campeche Sound, Gulf of Mexico, revealed by biomarker analysis, Organic Geochemistry, 39, 1104-1108, 2008.

Schubotz, F., Lipp, J. S., Elvert, M., and Hinrichs, K.-U.: Stable carbon isotopic compositions of intact polar lipids reveal complex carbon flow patterns among hydrocarbon degrading microbial communities at the Chapopote asphalt volcano, Geochimica et Cosmochimica Acta, 75, 4399-4415, 2011a.



Schubotz, F., Lipp, J. S., Elvert, M., Kasten, S., Mollar, X. P., Zabel, M., Bohrmann, G., and Hinrichs, K.-U.: Petroleum degradation and associated microbial signatures at the Chapopote asphalt volcano, Southern Gulf of Mexico, Geochimica et Cosmochimica Acta, 75, 4377-4398, 2011b.

Smith, A. J., Flemings, P. B., Liu, X., and Darnell, K.: The evolution of methane vents that pierce the hydrate stability zone in the world's oceans, Journal of Geophysical Research: Solid Earth, 119, 6337-6356, 2014.

Suess, E., Torres, M. E., Bohrmann, G., Collier, R. W., Rickert, D., Goldfinger, C., Linke, P., Heuser, A., Sahling, H., Heeschen, K., Jung, C., Nakamura, K., Greinert, J., Pfannkuche, O., Trehu, A., Klinkhammer, G., Whiticar, M. J., Eisenhauer, A., Teichert, B., and Elvert, M.: Sea floor methane hydrates at Hydrate Ridge, Cascadia Margin. In: Natural gas hydrates: Occurrence, distribution, and detection, Paull, C. (Ed.), Geophysical Monograph 124, American Geophysical Union, 87-98, 2001.

Teske, A. and Nelson, B. W.: The genera *Beggiatoa* and *Thioploca*, Prokaryotes, 6, 784-810, 2006.

Tissot, B. P. and Welte, D. H.: Petroleum formation and occurrence, Springer, Berlin, 699p, 1984.

Torres, M. E., Wallmann, K., Tréhu, A. M., Bohrmann, G., Borowski, W. S., and Tomaru, H.: Gas hydrate growth, methane transport, and chloride enrichment at the southern summit of Hydrate Ridge, Cascadia margin off Oregon, Earth and Planetary Science Letters, 226, 225-241, 2004.

Treude, T., Boetius, A., Knittel, K., Wallmann, K., and Jorgensen, B. B.: Anaerobic oxidation of methane above gas hydrates at Hydrate Ridge, NE Pacific Ocean, Marine Ecology Progress Series, 264, 1-14, 2003.

Valentine, D. L., Reddy, C. M., Farwell, C., Hill, T. M., Pizarro, O., Yoerger, D. R., Camilli, R., Nelson, R. K., Peacock, E. E., Bagby, S. C., Clarke, B. A., Roman, C. N., and Soloway, M.: Asphalt volcanoes as a potential source of methane to late Pleistocene coastal waters, Nature Geoscience, 3, 345-348, 2010.

Vernon, J. W. and Slater, R. A.: Submarine tar mounds, Santa Barbara County, Calinfornia, Bulletin of the American Association of Petroleum Geologists, 47, 1624-1627, 1963.

Weiland, R. J., Adams, G. P., McDonald, R. D., Rooney, T. C., and Wills, L. M.: Geological and biological relationships in the Puma appraisal area: From salt diapirism to chemosynthetic communities, Offshore Technology Conference, Houston, Texas, USA, 5-8 May 2008, OTC-19360-PP, 2008.

Whiticar, M. J.: A geochemial perspective of natural gas and atmospheric methane, Organic Geochemistry, 16, 531-547, 1990.

Williams, A. K., Lawrence, G. M., and King, M.: Exploring for deepwater petroleum systems with satellite SAR (Synthetic Aperture RADAR). Fact or Fiction? Comparing results from two of today's hotspots (Congo and Santos) with two of tomorrow's (Campeche and Cariaco) (2 Poster), Adapted from poster presentation of the AAPG Annual Convention, Houston, 2006 (available online: http://www.searchanddiscovery.com/documents/2006/06100williams/index.htm).

Williamson, S. C., Zois, N., and Hewitt, A. T.: Integrated site investigation of seafloor features and associated fauna, Shenzi field, deepwater Gulf of Mexico, Offshore Technology Conference, Houston, Texas, USA, 5-8 May 2008, OTC 19356, 2008.



**Figures**



**Figure 1.** Geomorphological setting of the southern Gulf of Mexico based on shaded GEBCO bathymetry. Campeche Knolls and Sigsbee Knolls are located within the sub-province 4S, which is part of the South Gulf Salt Province (outlined in blue) as suggested by Cruz-Mercado et al. (2011). Locations of definite (green dots) and probable (gray dots) oil slick origins at the sea surface according to Williams et al. (2006) and the extent of the respective study area (outer rectangle) are shown.



**Figure 2.** Swath bathymetry draped over GEBCO bathymetry of the Campeche Knolls and cruise track of M 114 (inset). Indicated are oil slick origins as inferred from oil slicks at the sea surface according to Williams et al. (2006) classified as definite (green dots) and probable (grey dots). Seafloor locations of hydrocarbon seepage sites investigated in this study are also shown (open circles, number in brackets). Specifics of those sites are given in Table 1.




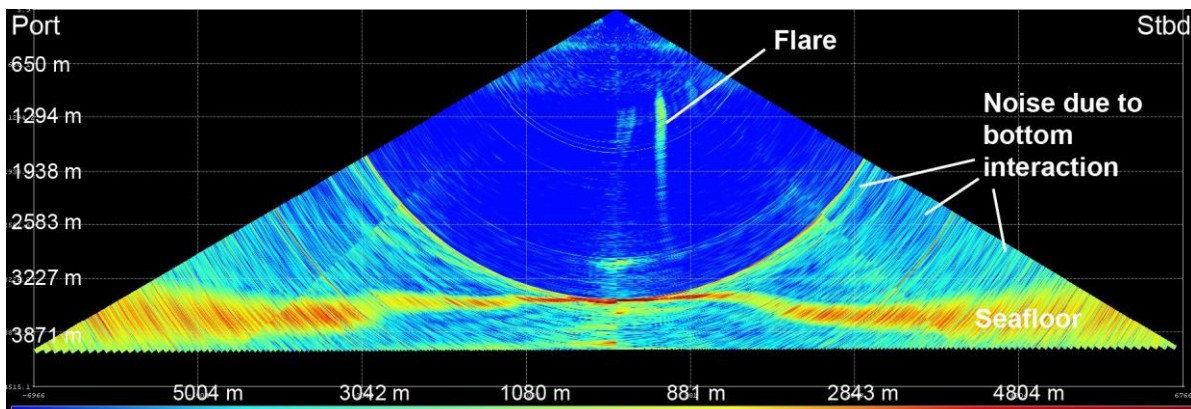

**Figure 3.** Screenshot of a multibeam swath showing a hydroacoustic flare that is caused by a plume of rising gas bubbles.




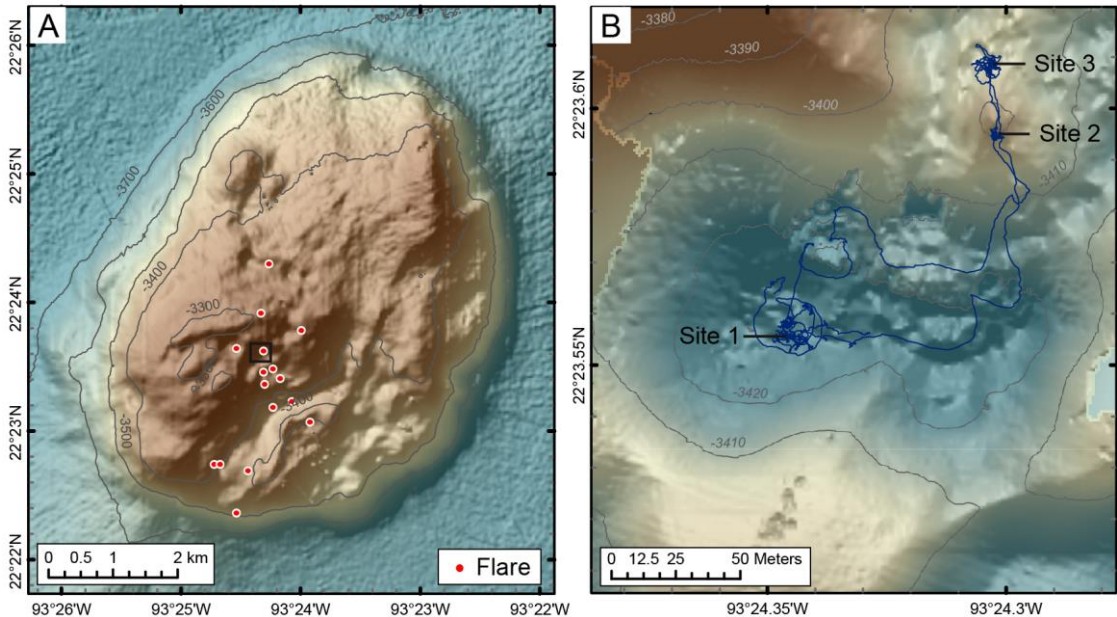

**Figure 4.** (A) Ship-based swath bathymetry of Tsanyao Yang Knoll, flare locations (red dots), and the area shown in (B) (box). (B) ROV QUEST Dive 361 track and the main study sites plotted on AUV swath bathymetry.





**Figure 5.** Seafloor images taken at Tsanyao Yang Knoll during ROV QUEST Dive 361. (A) Vestimentiferan tubeworm bushes on a dissected mound (arrow). Scale bar 50 cm. (B) Gas bubble plume (arrow) rose through the gap of a dissected mound. Hydrates formed at the hanging walls. Scale bar 50 cm. (C) Close-up of the mound surface percolated by bubbles (arrows). Gas hydrate occurred below a layer of authigenic carbonates with mytilids, vestimentifera, gastropods, and shrimps on top. Scale bar 10 cm. (D) Mound with outcropping gas hydrate (arrow) detailed in (E, F). Scale bar 50 cm. (E) Lens-shaped, outcropping gas hydrate composed of bubble-fabric hydrate below and dense hydrate above the arrow. Scale bar 50 cm. (F) Bubble-fabric hydrate inhabited by ice worms (cf. *Hesiocaeca methanicola*). Scale bar 1 cm. All images courtesy of MARUM.



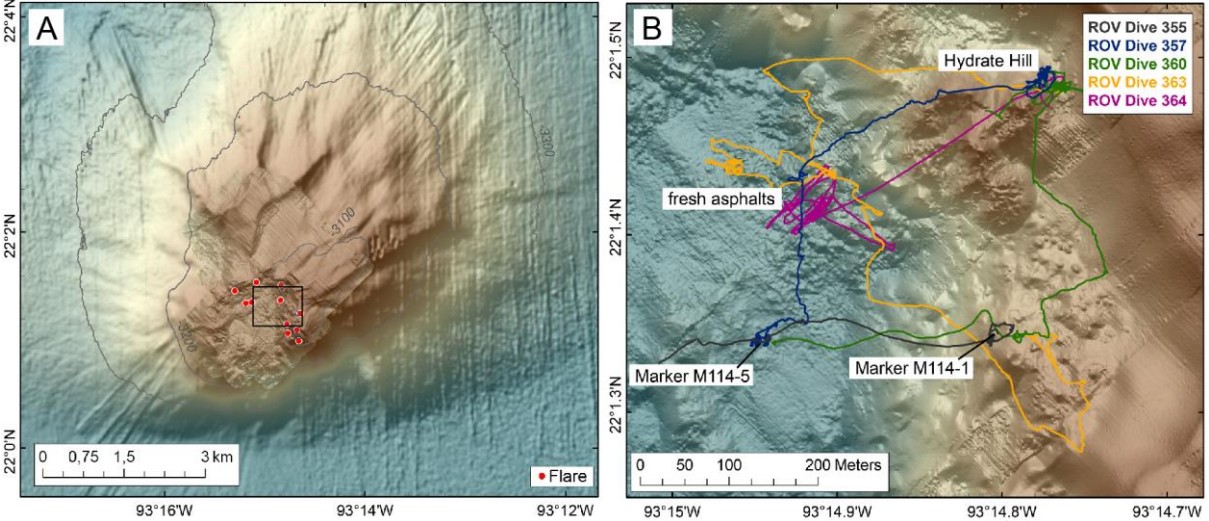

**Figure 6.** Maps of Mictlan Knoll: (A) AUV-based bathymetry draped over ship-based bathymetry and positions of flares (red dots). The box defines the area illustrated in (B). (B) ROV QUEST dive tracks and main study sites plotted on AUV-based bathymetry.



**Figure 7.** Seafloor images taken at Mictlan Knoll during ROV QUEST Dives 357 (A-D) and 360 (E-F). (A) Hydrate below overhanging vestimentiferan tubeworms (arrow). Scale bar 50 cm. (B) Gas bubble stream (arrow) rose through the hydrates and vestimentifera shown in (A). Scale bar 50 cm. (C) Fragmented asphalt, authigenic carbonates (arrow), and vestimentifera. Scale bar 50 cm. (D) Oil drops released through white-coated chimneys. Scale bar 20 cm. (E) Flourishing ecosystem (mytilids, vestimentifera with epizoic suspension feeders, bacterial mats) next to oil-soaked sediments shown in (F). (F) Viscous oil drops emanated from the sediments leaving strands behind (arrow). Scale bar 20 cm. All images courtesy of MARUM.



**Figure 8.** Seafloor images taken at Mictlan Knoll during ROV QUEST Dives 357 (B, C), 363 (A, E), and 364 (D-F). (A-C) Oil whips and sheets (arrows) floating in the water. Old whips and sheets apparently lost buoyancy and pile-up at the seafloor. (D-F) Flow structures of heavy oil. Scale bar all images 50 cm. All images courtesy of MARUM.




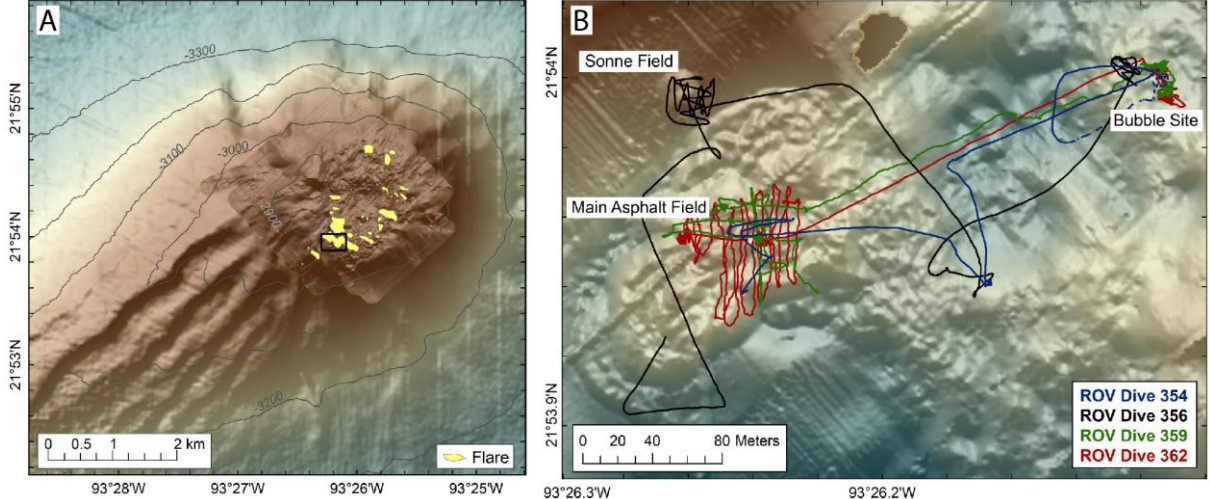

**Figure 9.** Maps of Chapopote Knoll: (A) AUV-based bathymetry draped over ship-based bathymetry and approximate positions of flares. The box shows the area in (B). (B) ROV QUEST dive tracks and main study sites plotted on AUV-based bathymetry.







**Figure 10.** Seafloor images taken at Chapopote Knoll during ROV QUEST Dives 354 (C, E), 362 (A, F), and 365 (B, D). (A) Main asphalt field with vestimentifera bordering a bacterial mat (arrow) on soft sediments. Scale bar 50 cm. (B) Fragmented asphalt. Scale bar 50 cm. (C) Catching hydrate-coated bubbles at the bubble site. Scale bar 10 cm. (D) Gas hydrate outcrop with mytilids, gastropods, galatheid crab. Note bubble fabric of exposed hydrate. Scale bar 50 cm. (E) Sponges, hydrozoans, mytilids at the bubble site. Scale bar 50 cm. (F) Oil drops and oil whips (arrows) close to the bubble site. Scale bar 50 cm. All images courtesy of MARUM.





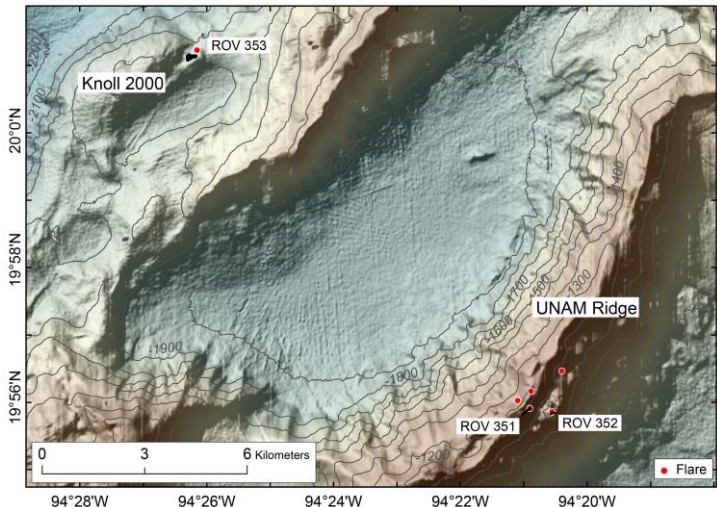

**Figure 11.** Ship-based bathymetry of Knoll 2000 and UNAM Ridge, locations of flares (red dots) and ROV QUEST dive tracks (black).





**Figure 12.** Seafloor images taken at UNAM Ridge (A-D) and Knoll 2000 (E-F) during ROV QUEST dives 352 and 353, respectively. (A) Soft coral and other suspension feeders on iron/manganese-stained authigenic carbonates. (B) Recumbent vestimentifera. (C) Authigenic carbonates and mytilid shells. (D) A few living mytilids attached to carbonates (foreground) and a 1 m wide circular depression. (E) Pogonophoran tubeworms were recovered from samples of whitish-stained sediments (arrow) next to carbonates inhabited by suspension feeders (*Actinoscyphia* sp., anemones, sponges). (F) Probably whitish bacterial mats on dark-stained sediments. Scale bar all images 50 cm. All images courtesy of MARUM.




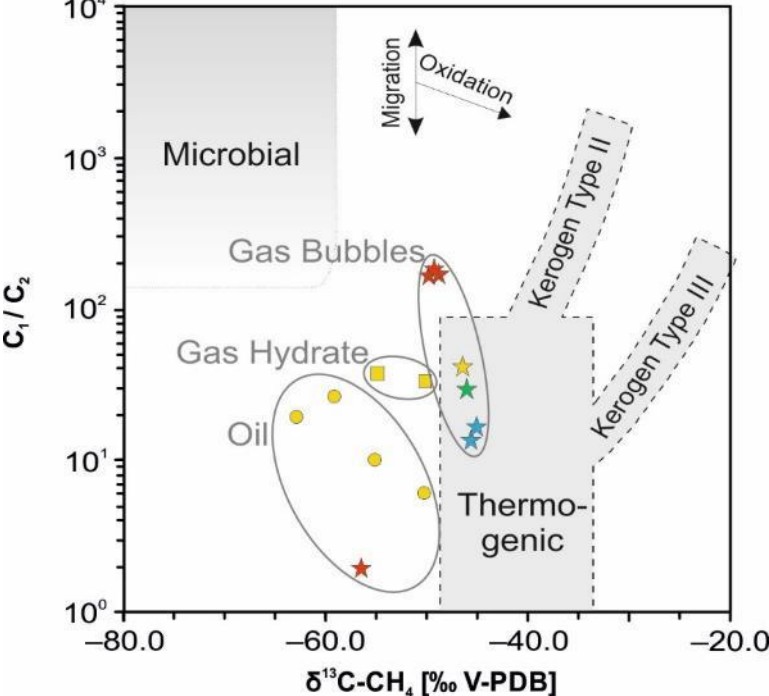

**Figure 13.** Molecular ($C_1/C_2$) vs. stable C isotopic composition of methane ($\delta^{13}C$-$CH_4$) sampled by Gas Bubble Sampler at Tsanyao Yang Knoll (blue), Mictlan Knoll (red), Chapopote Knoll (yellow), and UNAM Ridge (green) and a single oil-associated gas sample (Mictlan Knoll) collected during this study with the GBS. Stars indicate samples analyzed in this study, dots and squares are values according to results for methane in hydrates and oil collected during previous campaigns (MacDonald et al., 2004; Schubotz et al., 2011b). Classification according to the "Bernard diagram" modified after Whiticar (1990). Gas samples studied herein are plotted close to the empirical field of thermogenic methane.



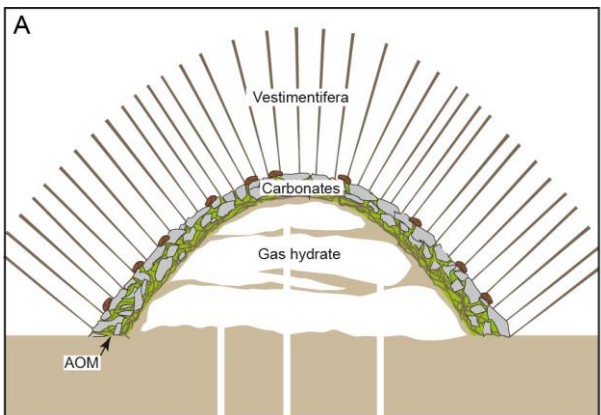
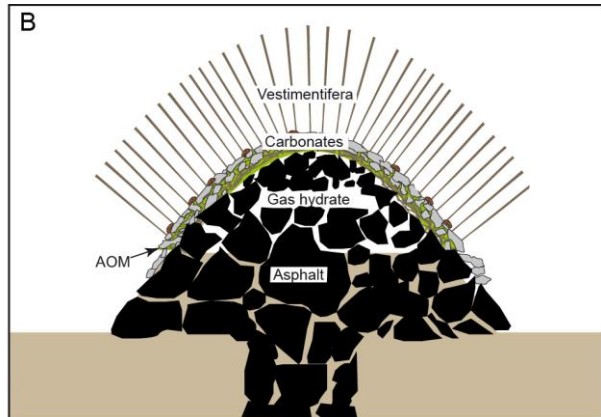

**Figure 14.** Sketch depicting the vestimentifera-gas/hydrate habitat encountered at (A) Tsanyao Yang Knoll and (B) Mictlan Knoll. Drawing not to scale. The mounds at Tsanyao Yang Knoll were a few meter wide while Hydrate Hill at Mictlan Knoll was about 30 m in diameter. AOM = anaerobic oxidation of methane (green).



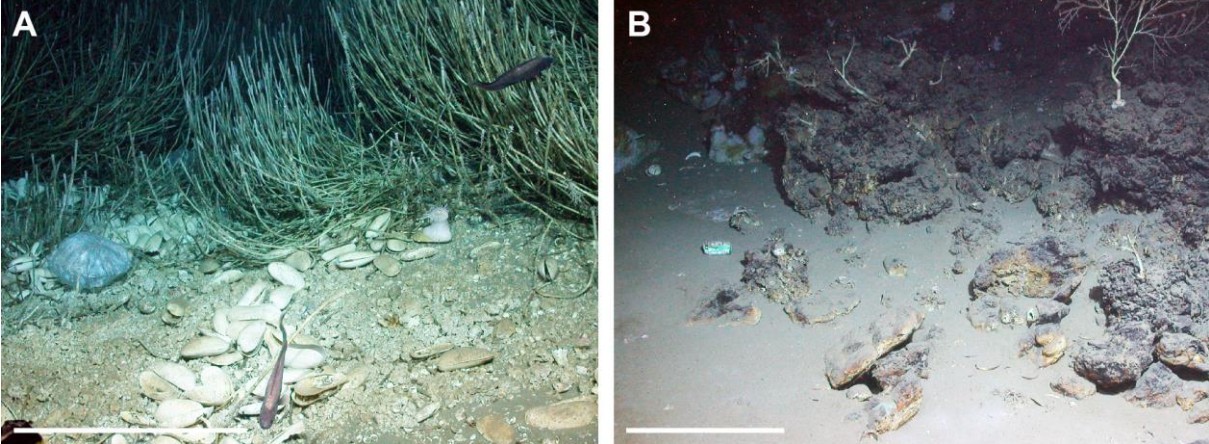

**Figure 15.** Seafloor images showing anthropogenic litter next to (A) vesicomyid clam *Abyssogena southwardae* at Tsanyao Yang Knoll and (B) carbonates at UNAM Ridge. Scale bar 50 cm. All images courtesy of MARUM.





## Tables

**Table 1.** Summary of ROV and camera-sled observations of evidence for hydrocarbon seepage in the southern Gulf of Mexico obtained during cruises R/V *Sonne* cruise 174, R/V *Meteor* cruise 67/2, R/V *Justo Sierra* cruise Campeche III, and R/V *Meteor* cruise 114/2.

| Seafloor Structure | Location | Tools | Observation |
|---|---|---|---|
| 1 | 20°4.95'N; 93°57.85'W, 1300 m | R/V *Justo Sierra* camera survey, Site L89 | Asphalt, vestimentifera, soft coral |
| 2 | 20°19.55'N; 93°59.15'W; 1780 m | R/V *Justo Sierra* camera survey, Site L87 | Asphalt or carbonate pieces, sparse bacterial mats |
| 3 | 20°22.02'N; 93°54.28'W; 1700 m | R/V *Justo Sierra* camera survey, Site L86 | Asphalt, tubeworm, mussels |
| 4 | 21°11.45'N; 93°53.40'W; 2335 m | R/V *Justo Sierra* camera survey, Site L97 | Asphalt or carbonate |
| 5 | 19°54.917'N; 93°56.366'W; 1100 m | R/V Justo Sierra camera survey, Site L94 | Asphalt or carbonate, bacterial mats |
| 6 | 21°39.5'N; 93°26.1'W, 2980 m | M 67/2 TV-Sled survey, Knoll 2139 | Asphalt deposits some ten meters in diameter |
| 7 | Ca. 21°25'N; 93°22'W; 2400 m and 21°23.8'N; 93°23.3'W; 2440 | SO 174/2 TV-Sled (OFOS 10 and 12), Knoll 2124 | Sediment covered, outcropping asphalts, scattered living vesicomyid clams, vestimentifera, bacterial mats |
| Tsanyao Yang Knoll | 22°23.55'N; 93°24.33'W, 3420 m | M 114/2 ROV Dives 358, 361 | Vestimentifera on hydrate outcrops, gas bubble and oil emission, carbonates, mytilid and vesicomyid bivalves |
| Mictlan Knoll | 22°1.4'N; 93°14.9'W; 3180 m | M 114/2 ROV Dives 355, 357, 360, 363, 364 | Asphalt deposits, gas bubble and oil emission, vestimentifera, white bacterial mats, mytilid and vesicomyid bivalves |
| Chapopote Knoll | 21°53.95'N; 93°26.25'W; 2920 m | SO 174 OFOS 13 and 14; M 67/2 ROV Dives 80-84; M 114/2 ROV Dives 354, 356, 359, 363 | Asphalt deposits, gas bubble and oil emission, vestimentifera, white bacterial mats, mytilid and vesicomyid bivalves |
| UNAM Ridge | 19°55.90'N; 94°20.89'W; 1230 m | M 114/2 ROV Dives 351, 352 | Asphalt deposits, carbonates, vestimentifera, mytilids, gas bubble emission |
| Knoll 2000 | 20°01.12'N; 94°26.20'W; 1860 m | M114/2 ROV Dive 353 | Carbonates and possibly asphalts, white bacterial mats on blackish seafloor, pogonophoran tubeworms |