# Peer review of "Massive asphalt deposits, oil seepage, and gas venting support"

_Biogeosciences, 2016_

## Short Comment (SC1) · 4 Apr 2016

One application of these results is consideration of seafloor hazards as the exploration program now operating in the Campeche offshore gathers momentum. Seismic and bathymetric surveys are unlikely to have impacts, but piston coring & heat-flow probes, among other operations, could locally impact sensitive chemosynthetic communities and potentially cause release of shallow gas and oil pools. It would be useful to give some priority in terms of backscatter, bathymetric, or other indicators that might be associated with the described sites. This would expand/refine the discussion on pg. 17 and point to some of the results from the individual knolls. At present, hazard avoidance would have to be based on a combination of plume detection and some

rather imprecise interpretation of the bathymetry–avoiding crater walls for example. Possibly some backscatter data would improve this.

---

## Referee Comment (RC1) · Anonymous Referee #1 · 10 May 2016

This manuscript reports a descriptive study of surveys and underwater AUV and ROV observations at the Campeche Knolls region in the southern GoM. It includes highly detailed but non-quantitative descriptions of the oil and gas based discharges, seabed structure, and biological communities. This manuscript describes an interesting location. This would be acceptable for publication with minor to moderate revisions.

My main points are that the abstract needs to be shortened and made more concise. And there are some aspects of the discussion, particularly related to the biological communities, should be modified according to my specific comments below. Additionally, I suggest some minor grammatical modifications. My specific comments follow below:

Title: No mention of the chemosynthetic fauna, yet they are a central part of the results and discussion. Why?

Abstract: It strikes me as too detailed. It needs to be streamlined to summarize the key points, and lose some of the detail. Also, the order of topics (method employed to data acquisition, habitat, community, gas composition, gas emission, hydrate and fauna, summary) is somewhat chaotic and could be reordered and better integrated to make the information smoother. Also the part of the final sentence on "species new to science" is unsubstantiated, not discussed elsewhere in the paper and should be deleted.

Introduction: para 3: "scuba-diving depths" is subjective. What is it, 30m?

Introduction: para 3, last sentence: on oil exploration. This sentence is not really part of the paragraph or the paper and should be eliminated, or developed more fully.

Methods, para 2: Please provide the exact dates for cruise M-114. Same for Table 1 (see comment below).

Results 4.1: Paragraph 2. The backscatter profiles that you show, and the situation you discuss related to its appearance in only part of the water column can also be due to currents. If this is the case, then linearly projecting the flare from mid-water to the seabed may be biased. You might want to at least mention this.

Site Description 4.2.1: replace "in the following" with 'hereafter'

4.2.2. The term "decimeter" while not incorrect, seems somewhat awkward to me. Perhaps consider replacing it with or tens of cm??? Sorry if this seems nitpicky, and I am certainly willing to yield to the editor on this if we differ in opinion.

4.2.3. In the mention of the bivalves and other fauna found here, was there any collection made for analysis of these organisms?

4.3. Were the camera sled observations from previous cruises included in this paper?
It seems ambiguous.

Discussion, 5.1, para 2, sentence 1. "While prior..." Awkward wording. Reformulate.

Discussion, 5.1, para 3, sentence 1. Camera sled surveys. It is unclear to me what data the camera sled surveys have added to the present manuscript.

Discussion, 5.1, para 2, sentence 1: "proven for". Revise as "visually identified at"

Sec 5.3: find and fix spelling error "alcalinity"

Sec 5.3. "...we speculate that gas seepage at our study sites was stable on time scales of hundreds of years...". Be careful here. I see your point and will not wholly disagree, but the only chronometer you are invoking are vestimentiferan "estimated" lifespans from a completely different location. Further the Bergquist method of aging was not unequivocal, so be a little careful here.

Sec 5.5: many mentions of bacteria: "methane oxidizing bacteria", chemosynthetic bacteria". Unless the authors are sure, these microbes could well be archaea. I suggest using "microbes" instead, or specify if they are archaea or bacteria.

Sec 5.5: (final paragraph): "Preliminary interpretation of our observations suggest that the species diversity is higher in the oil seeps that at other sites..." This statement is premature and unsupportable in its present form. If the authors think that this may be the case, it would be easy enough to quantify with a proper analysis of species present and their abundances. Either do a proper analysis, or drop this statement.

Sec 5.6: This entire section has several problems, and could, in my opinion, be eliminated. It really is reaching outside the core story and does not add to the central thesis of the paper.

The first paragraph on biogeography and teleconnections between Campeche Knolls and other deep-water seep systems, particularly the Florida escarpment, is very speculative, and based on only the thinnest of observations from this study. In fact, as mentioned earlier in my review, the analysis of the benthic community, species present, and community structure, and diversity is not very well developed. A lot more formal analysis could be made of the observations of the community characteristics. Lacking that, this paragraph is unwarranted.

The second paragraph on anthropogenic impacts of the benthic community is not germane to the story and can be eliminated in its entirety.

The final paragraph on advocating for a priori protection of these locations in any future oil exploration is really advocacy, and not basic science. In my opinion, this is out of scope and should be deleted (the parts of the abstract and conclusions regarding this should also be modified accordingly).

Conclusion:

Typo: 'reanalyzes' = reanalysis

Last sentence, first para: Delete this sentence. Not a main conclusion, and no direct supporting evidence.

Last sentence, second para: "over time spans of hundreds of years". Really no direct evidence for this. Change to "over extended timespans"

Last sentence, final paragraph: "We call for protective..." Advocacy. Delete (see also comment above).

Figure 4A. There is no box shown.

Figure 9A. There is no box shown.

Figure 11. Cannot see the ROV dive tracks. Possibly the image is too dark.

Table 1. Put the stations from the current cruise first (not last). Also add dates for the AUV or ROV dives or other observations. Also, place depth of each location in a separate column.

[Figure]

---

## Referee Comment (RC2) · Anonymous Referee #2 · 13 May 2016

This manuscript brings much needed exploratory work to the study of asphalt seeps in the southern Gulf of Mexico. Previous research in this area has focused solely on Chapopote Knoll. This study expanded into 11 additional sites in the Campeche Knolls to find widespread evidence of asphalt seepage in the region and provided detailed site descriptions of the geology and biology found at these new locations. This manuscript is thorough in laying the foundation for future deep-sea seep research at these newly-explored sites, and should be accepted for publication after minor revisions.

Comments:

1. I agree with Reviewer 1 that it seems incongruous for the title to have no mention of the chemosynthetic communities discussed in this study.

[Figure]

2. The Abstract is too long and detailed to give the reader a concise snapshot of the study and should be condensed from three paragraphs to one.

3. It is odd that the tubeworms, which are frequently mentioned in the text and correctly identified as vestimentiferans, are not more specifically called Escarpia sp. until page 17 (late in the Discussion section). The depth at which these tubeworms were found combined with the genetic identification from Raggi et al. 2013 (cited in the manuscript) support the use of this genus in the manuscript. Several other common chemosynthetic megafauna are identified by species name in the text (e.g. Bathymodiolus brooksi, B. heckerae, and Abyssogena southwardae), so it is incongruous for the tubeworms to be identified by "vestimentifera" only.

4. Additionally, this means that the Campeche Knolls tubeworms are definitely a different species from Lamellibrachia luymesi, the species whose age was estimated in Bergquist et al. 2000. That study of the northern GoM species is cited here to estimate that the vestimentiferan-inhabited asphalt flows found in this study could potentially be decades old. The last paragraph of section 5.3 should more accurately state the species discrepancy (they are not merely "likely" a different species from the northern GoM study) and show caution in using this age estimate.

5. The Results section 4.1 "Gas emissions from the seafloor" may be better incorporated into the manuscript as part of the Methods section. This subsection does describe the results of the multi-beam echosounder surveys, but more importantly it describes how the authors used this information to trace the origin of bubble flares and choose sites for more in-depth AUV and ROV surveys. It then logically follows that the site descriptions and gas bubble samples obtained from those video surveys that make up the rest of the Results section were direct results of this decision-making process.

6. Figure 1 is very helpful in displaying different features of the southern Gulf of Mexico, but the gray and green dots meant to represent probable and definite seeps respectively are hard to distinguish. Although this color scheme is easier to differentiate when

the area is magnified in Figure 2, the sites would be better served with different color choices.

7. Figure 3 is clear, but ultimately doesn't contribute much to the manuscript. The text description of identifying gas bubble plumes from multibeam echosounder seems sufficient to communicate the methods of the study to the reader and explain that plumes were not always traceable to the seafloor.

8. The dark blue box in Figure 4A showing the ROV survey area is difficult to distinguish from the background bathymetry.

Typographical errors:

- Last sentence of first paragraph of Introduction: "bolder" should be corrected to "boulder."

- Same issue in second paragraph of section 4.2.1 ("bolder" instead of "boulder")

- Last paragraph of section 4.2.2 (bottom of page 8): "loose buoyancy" should be corrected to "lose buoyancy."

- Last paragraph of section 5.1: "temporarily and spatially segregated" should be corrected to "temporally and spatially segregated."

- Last paragraph of section 5.3: I believe the authors meant "slow growth" rather than "low growth."

- First paragraph of section 5.4: Mictlan Knoll is misspelled as "Mictan Knoll" in the first sentence, and in the third sentence API gravity should be "slightly higher" rather than "slighter higher."

---

## Short Comment (SC2) · Sahling et al · 19 May 2016

See comments and edits in text and figures (scanned as separate .pdf files; attached).

Fig. 1. scanned text with edits and comments

**Fig. 2.** scanned figures with edits and comments

---

## Short Comment (SC3) · 21 May 2016

Attached you will hopefully find a document with the text and figures commented upon, with a few minor corrections.

A few general / high-level comments:

See comment on page 5: We suggest adding a sentence to clarify what the authors mean by "flares", "plumes" and water column anomalies. Something like, "In this paper we assume that all acoustic flares are related to gas seepage and we will refer to them as "gas bubble flares", "flares" or "plumes"

Page 6: please explain whether the "flare" picks are the lowest actual pick you could

make in the water column, whether it's the projected seafloor point of origination using the angle of the identified flare in the water column, or if it is vertically below the deepest identified flare point.

Page 11: Discussion. Did you study any regions without SAR slicks? If not, then you've established a correlation between SAR slicks and seafloor seepage, but the converse – whether there's seepage in the absence of SAR slicks – has not been shown. This is the "evidence of absence" that philosophers refer to. The authors should sate this clearly. By studying only areas with SAR slicks, there is a sample bias. This isn't a serious issue, but it does need to be stated.

Page 17: If the 11 SAR sites studied all had evidence of seepage, and there are over 50 SAR sites. . .how unique is the ecosystem? Each of the knolls studied cover on the order of 50 sq. km. or more. The very small detailed areas on each knoll showed evidence of seepage. Could this evidence of seepage extend over a large area of each knoll? On a worldwide basis, this type of ecosystem is rare, but in the Campeche area is it possible that it is not – that it is actually common?

Page 17 – 18: the authors need to significantly expand their discussion of the oil industry and potential impacts on seep communities. Specifically, since the first cold seep communities were discovered in 1985 (including the Gulf of Mexico), the recognition of seep communities in an area of active exploration and development lead to a series of "notices to lessses" regarding how to avoid biologically sensitive areas. See NTL 2009-G40 (link: http://www.boem.gov/Regulations/Notices-To-Lessees/2009/09-G40.aspx ). The NTL provided guidance on how to develop in an area of potential seep communities (and deepwater corals). Geohazard interpreters have gain experience interpreting potential seep locations based upon geophysical data, and these geohazard surveys are a routine part of exploration in the northern Gulf of Mexico. The authors' work will in fact help provide a foundation for where such communities are found in the southern Gulf of Mexico, and a basis for an approach similar to BOEM's might be applied to the Mexican Gulf of Mexico.

As the authors will see on the bottom of page 18, we suggest they consider the addition of a sentence such as "We call for the impact on these ecosystems to be considered as part of any future development in the Campeche Knolls area."

I strongly recommend publication with minor modification.

Respectfully submitted, -Dan Orange ONE / U.C. Santa Cruz

Please also note the supplement to this comment:
http://www.biogeosciences-discuss.net/bg-2016-101/bg-2016-101-SC3-supplement.pdf

––––––––––––––––––––––––

**Supplement:**

[revised manuscript text omitted]

---

## Referee Comment (RC3) · Anonymous Referee #3 · 25 May 2016

The Manuscript by Sahling and colleagues about the Campeche Knolls area of asphalt deposits describes a series of deep-sea reducing habitats that have similarities and yet distinct features compared to those known. While the manuscript is largely descriptive, its careful inclusion of these sites in current knowledge of reducing habitats and appropriate interpretation of its results provides a useful addition to our overall understanding of the diversity of habitats found in the deep sea. In this particular case it also adds to a section of the planet that is largely unknown and yet lies in close proximity to among the best studied regions (the GoM) for similar habitats on the globe. Further, the exciting observations of the vestimentiferan 'rhizosphere' community was an intriguing discovery and provides the opportunity for conceptual advancement. This manuscript

was carefully and well written and thus I have only very minor comments and feel this will be a valuable addition to the literature and is appropriate for Biogeosciences.

The introduction, methods and results were all well written and of appropriate length and depth. The authors demonstrated a clear grasp of the pertinent literature. The discussion nicely placed these results in correct context and then used these to highlight the knowns and unknowns of the regions. Imagery was used powerfully to illustrate the discoveries and I can see that many of these figures will be used in the future in discussions of the different types of reducing habitats in the deep sea.

I have the following minor comments that would marginally improve the manuscript: 4.2.4 – 'pogonophoran' has fallen out of favor and I am unfamiliar with Anobturata and it is not in WORMs. Can the authors either provide a reference for Anobturata or use frenulate or monolifera (depending on which it may be – with frenulate the more likely of the two and currently more 'correct' (i.e. monophyletic) compared to pogonophoran.) P13 – "we sketched" suggest "we illustrate" P14 - "As well as oil derived from higher hydrocarbons" – I would suggest changing this to "and likely augmented by degeneration of organic carbon that may include higher hydrocarbons". As Joye et al. found a mismatch between SRR and AOM. They did not completely contribute the sulfide present to longer chain hydrocarbons. P14 "With estimated life spans" – additional age info is found in Cordes et al. 2007 Marine Ecology 28:160-168 that puts smaller individuals (1.1m) at age estimates of 300 years. This provides further evidence for the long term seepage estimates provided by the authors.
* * *

---

## Author Comment (AC1) · 3 Jun 2016

Thank you for indicating that seafloor hazards due to the exploration program should be considered. We agree that additional geophysical data such as backscatter would help avoiding disturbance of the chemosynthetic communities. The information is available and part of an ongoing study to interpret the geophysical data. It is planned to publish these results in a follow-up publication.

---

## Author Comment (AC2) · 3 Jun 2016

We thank Referee #1 for the thorough revision of our manuscript that helped to make it more concise. We followed all of the referee's recommendations and applied changes to the manuscript accordingly. A revised version of the manuscript considering the comments of Referee #1, #2, and #3 as well as the SC3 has been uploaded, including a version with tracked changes.

We shortened the abstract, modified the title, deleted Section 5.6., and omitted Figure 15 which was part of Section 5.6. All of the ambiguous wordings have been rephrased and all spelling mistakes corrected. We re-phrased the paragraph concerning the use of vestimentifera as chronometer and are more careful about its validity. The dates of

the ships cruise and the station work were included in the text and Table 1. The figures were redrawn.

There are two comments by the referee that we considered and applied changes for making the points more clear that we would like to comment explicitly here:

Following the recommendations by Referee #2 we moved and integrated the section Results 4.1 into the Material and Method section. We describe in more detail how flares were traced through the water column analyzing swath by swath manually. Such three dimensional analyses allows to trace the flares through the water column although deviated by currents. We are therefore confident that the fact that flares only appeared above the seafloor in the echosounder records are not due to currents, as suggested by Referee #1.

Referee #1 questions the benefit of the camera sled surveys to the present manuscript. The camera sled surveys as summarized in Table 1 double the locations with evidence for hydrocarbon seepage at Campeche Knolls as shown in Figure 2. It supports the fact that this particular form of hydrocarbon seepage with asphalt deposits at the seafloor is not limited to the sites described in detail by ROV but are more wide spread and, thus, an integral component of seepage at Campeche Knolls. The importance of such observation has also be emphasized by Referee #2 that is why we left these information integrated into the manuscript.

---

## Author Comment (AC3) · 3 Jun 2016

We thank Referee #2 for the important comments and considered all, which improved the sharpness and readability of the manuscript. A revised version of the manuscript considering the comments of Referee #1, #2, and #3 as well as the SC3 has been uploaded, including a version with tracked changes.

Explicitly, we have re-written and shortened the abstract. We re-phrased the title to: "Massive asphalt deposits, oil seepage, gas venting support abundant chemosynthetic communities at Campeche Knolls, southern Gulf of Mexico". Section 4.1 was integrated into the section Material and Methods as recommended by Referee #2 and Fig. 3 omitted. The figures were redrawn and all typographic errors corrected.

[Figure]

We agree with the referee that it is likely that we encountered the genus Escarpia in our study area and applied changes accordingly. However, we found several morphotypes of vestimentifera in our study and feel like being careful with ascribing all to that genus Escarpia until morphological or phylogenetic studies confirm this. Therefore, we did leave the more general description vestimentifera in several instances. We rephrased the paragraph concerning the use of vestimentifera as chronometer and are more careful about its validity.

———————————————————

---

## Author Comment (AC4) · 3 Jun 2016

Thanks a lot for the constructive suggestions and comments. We applied the suggested changes and corrected almost all of your editorial comments as written in the attached pdf file you uploaded. A revised version of the manuscript considering your comments as well as those of Referee #1, #2, and #3 has been uploaded, including a version with tracked changes.

Thanks for your comment on the correlation between SAR slicks and seafloor seepage and your note on "evidence of absence" that we fully agree with. We applied changes accordingly.

[Figure]

Three of your general / high-level comments concerned the former section 5.6 "Chemosynthetic communities at Campeche Knolls and future energy exploration" as well as a sentence concerning this topic in the section Conclusion. We acknowledge your comments on regulations regarding how to avoid biological sensitive areas (e.g. the "notices to lessees"). However, we deleted the respective sections from the manuscript following the advice by Referee #1 as this section is "advocacy and not basic science", which we agree on.
* * *

---

## Author Comment (AC5) · 3 Jun 2016

We thank Referee #3 for the positive revision. We applied the suggested changes. However, as two reviewers suggested to be more cautious in the age estimates based on vestimentiferan tube length, we included the reference by Cordes et al. (2007) as suggested by Referee #3 but changed the discussion following Referee #1 and #2.

A revised version of the manuscript considering the comments of Referee #1, #2, and #3 as well as the SC3 has been uploaded, including a version with tracked changes.

---

## Author Comment (AC6) · 3 Jun 2016

We thank all the Referees and Danial Orange for their constructive suggestions and comments, which helped to improve the manuscript. Almost all of the comments have been applied (see the individual replies).

We are grateful for the comments by all referees and included a sentence in the section Acknowledgments.

After submission of the original version of the manuscript, we became aware of two mistakes that we corrected:

The sequence of co-authors was not correct but now follows the alphabetic order be-

tween the first and last authors.

We integrated a reference to the PhD thesis of Gopika Suresh and thank her for her ambitious work in the section Acknowledgments.

We hope, with applying the modification to the manuscript, it is now acceptable for publication.

Please also note the supplement to this comment:
http://www.biogeosciences-discuss.net/bg-2016-101/bg-2016-101-AC6-
supplement.zip

---

## Author Comment (AC7) · 29 Jun 2016

Point-by-point comment to Referee #1

Referee #1: My main points are that the abstract needs to be shortened and made more concise. And there are some aspects of the discussion, particularly related to the biological communities, should be modified according to my specific comments below. Additionally, I suggest some minor grammatical modifications. My specific comments follow below: Title: No mention of the chemosynthetic fauna, yet they are a central part of the results and discussion. Why?

Reply: We changed the title to: "Massive asphalt deposits, oil seepage, and gas venting support abundant chemosynthetic communities at Campeche Knolls, southern Gulf of Mexico"

Referee #1: Abstract: It strikes me as too detailed. It needs to be streamlined to summarize the key points, and lose some of the detail. Also, the order of topics (method employed to data acquisition, habitat, community, gas composition, gas emission, hydrate and fauna, summary) is somewhat chaotic and could be reordered and better integrated to make the information smoother. Also the part of the final sentence on "species new to science" is unsubstantiated, not discussed elsewhere in the paper and should be deleted.

Reply: We have rewritten the abstract following the recommendations.

Referee #1: Introduction: para 3: "scuba-diving depths" is subjective. What is it, 30m?

Reply: Correct, it was at 30 m water depth, we changed the sentence accordingly.

Referee #1: Introduction: para 3, last sentence: on oil exploration. This sentence is not really part of the paragraph or the paper and should be eliminated, or developed more fully.

Reply: We deleted the sentence.

Referee #1: Methods, para 2: Please provide the exact dates for cruise M-114. Same for Table 1 (see comment below).

Reply: We included the dates of the cruise in the text and included the dates of all stations conducted in Table 1.

Referee #1: Results 4.1: Paragraph 2. The backscatter profiles that you show, and the situation you discuss related to its appearance in only part of the water column can also be due to currents. If this is the case, then linearly projecting the flare from mid-water to the seabed may be biased. You might want to at least mention this.

Reply: We have re-written the paragraph and now describe in more detail how flares

were traced through the water column analyzing swath by swath manually. Such three dimensional analyses allows to trace the flares through the water column although deviated by currents. We are therefore confident that the fact that flares only appeared above the seafloor in the echosounder records are not due to currents.

Referee #1: Site Description 4.2.1: replace "in the following" with 'hereafter'

Reply: Done.

Referee #1: 4.2.2. The term "decimeter" while not incorrect, seems somewhat awkward to me. Perhaps consider replacing it with or tens of cm??? Sorry if this seems nitpicky, and I am certainly willing to yield to the editor on this if we differ in opinion.

Reply: We changed the phrase giving the more accurate numbers "10 to 30 cm in height".

Referee #1: 4.2.3. In the mention of the bivalves and other fauna found here, was there any collection made for analysis of these organisms?

Reply: We included two sentences in the Material and Method section indicating that biological samples were taken but that the taxonomic identification is far from being complete.

Referee #1: 4.3. Were the camera sled observations from previous cruises included in this paper? It seems ambiguous.

Reply: Yes, we used camera sled observations from previous cruises and included the information explicitly into the result section.

Referee #1: Discussion, 5.1, para 2, sentence 1. "While prior: : :" Awkward wording. Reformulate.

Reply: Done.

Referee #1: Discussion, 5.1, para 3, sentence 1. Camera sled surveys. It is unclear to

me what data the camera sled surveys have added to the present manuscript.

Reply: The camera sled surveys as summarized in Table 1 double the locations with evidence for hydrocarbon seepage at Campeche Knolls as shown in Figure 2. It supports the fact that this particular form of hydrocarbon seepage with asphalt deposits at the seafloor is not limited to the sites described in detail by ROV but are more wide spread and, thus, an integral component of seepage at Campeche Knolls. The importance of such observation has also be emphasized by Referee #2 that is why we left these information integrated into the manuscript.

Referee #1: Discussion, 5.1, para 2, sentence 1: "proven for". Revise as "visually identified at"

Reply: Done.

Referee #1: Sec 5.3: find and fix spelling error "alcalinity"

Reply: Done.

Referee #1: Sec 5.3. ": : :we speculate that gas seepage at our study sites was stable on time scales of hundreds of years: : :". Be careful here. I see your point and will not wholly disagree, but the only chronometer you are invoking are vestimentiferan "estimated" lifespans from a completely different location. Further the Bergquist method of aging was not unequivocal, so be a little careful here.

Reply: We re-phrased the paragraph concerning the use of vestimentifera as chronometer and are more careful about its validity.

Referee #1: Sec 5.5: many mentions of bacteria: "methane oxidizing bacteria", chemosynthetic bacteria". Unless the authors are sure, these microbes could well be archaea. I suggest using "microbes" instead, or specify if they are archaea or bacteria.

Reply: We refer to publications that have studied the microbes and identified them as bacteria, so we left the section unchanged.

[Figure]

Referee #1: Sec 5.5: (final paragraph): "Preliminary interpretation of our observations suggest that the species diversity is higher in the oil seeps that at other sites..." This statement is premature and unsupportable in its present form. If the authors think that this may be the case, it would be easy enough to quantify with a proper analysis of species present and their abundances. Either do a proper analysis, or drop this statement.

Reply: We omitted the statement.

Referee #1: Sec 5.6: This entire section has several problems, and could, in my opinion, be eliminated. It really is reaching outside the core story and does not add to the central thesis of the paper. The first paragraph on biogeography and teleconnections between Campeche Knolls and other deep-water seep systems, particularly the Florida escarpment, is very speculative, and based on only the thinnest of observations from this study. In fact, as mentioned earlier in my review, the analysis of the benthic community, species present, and community structure, and diversity is not very well developed. A lot more formal analysis could be made of the observations of the community characteristics. Lacking that, this paragraph is unwarranted. The second paragraph on anthropogenic impacts of the benthic community is not germane to the story and can be eliminated in its entirety. The final paragraph on advocating for a priori protection of these locations in any future oil exploration is really advocacy, and not basic science. In my opinion, this is out of scope and should be deleted (the parts of the abstract and conclusions regarding this should also be modified accordingly).

Reply: We agree that the paragraph has several problems and deleted the entire section.

Referee #1: Conclusion: Typo: 'reanalyzes' = reanalysis

Reply: Done.

Referee #1: Last sentence, first para: Delete this sentence. Not a main conclusion,

and no direct supporting evidence.

Reply: Done.

Referee #1: Last sentence, second para: "over time spans of hundreds of years". Really no direct evidence for this. Change to "over extended timespans"

Reply: Done.

Referee #1: Last sentence, final paragraph: "We call for protective: : :" Advocacy. Delete (see also comment above).

Reply: Done.

Referee #1: Figure 4A. There is no box shown.

Reply: We changed the color of the box to become more visible.

Referee #1: Figure 9A. There is no box shown.

Reply: We changed the color of the box to become more visible.

Referee #1: Figure 11. Cannot see the ROV dive tracks. Possibly the image is too dark.

Reply: We changed the color of the dive tracks to become more visible.

Referee #1: Table 1. Put the stations from the current cruise first (not last). Also add dates for the AUV or ROV dives or other observations. Also, place depth of each location in a separate column.

Reply: Done.

---

## Author Comment (AC8) · 29 Jun 2016

Point-by-point reply to Referee #2

Referee #2: 1. I agree with Reviewer 1 that it seems incongruous for the title to have no mention of the chemosynthetic communities discussed in this study.

Reply: We changed the title to "Massive asphalt deposits, oil seepage, gas venting support abundant chemosynthetic communities at Campeche Knolls, southern Gulf of Mexico".

Referee #2: 2. The Abstract is too long and detailed to give the reader a concise snapshot of the study and should be condensed from three paragraphs to one.

[Figure]

Reply: We have rewritten the abstract following the recommendations.

Referee #2: 3. It is odd that the tubeworms, which are frequently mentioned in the text and correctlyidentified as vestimentiferans, are not more specifically called Escarpia sp. until page 17 (late in the Discussion section). The depth at which these tubeworms were found combined with the genetic identification from Raggi et al. 2013 (cited in the manuscript) support the use of this genus in the manuscript. Several other common chemosynthetic megafauna are identified by species name in the text (e.g. Bathymodiolus brooksi, B. heckerae, and Abyssogena southwardae), so it is incongruous for the tubeworms to be identified by "vestimentifera" only.

Reply: We agree with the referee that it is likely that we encountered the genus Escarpia in our study area and applied changes accordingly. However, we found several morphotypes of vestimentifera in our study and feel like being careful with ascribing all to that genus Escarpia until morphological or phylogenetic studies confirm this. Therefore, we did leave the more general description vestimentifera in several instances.

Referee #2: 4. Additionally, this means that the Campeche Knolls tubeworms are definitely a different species from Lamellibrachia luymesi, the species whose age was estimated in Bergquist et al. 2000. That study of the northern GoM species is cited here to estimate that the vestimentiferan-inhabited asphalt flows found in this study could potentially be decades old. The last paragraph of section 5.3 should more accurately state the species discrepancy (they are not merely "likely" a different species from the northern GoM study) and show caution in using this age estimate.

Reply: We agree and re-phrased the paragraph accordingly. We are more careful about the validity concerning the use of vestimentifera as chronometer.

Referee #2: 5. The Results section 4.1 "Gas emissions from the seafloor" may be better incorporated into the manuscript as part of the Methods section. This subsection does describe the results of the multi-beam echosounder surveys, but more importantly it describes how the authors used this information to trace the origin of bubble flares

and choose sites for more in-depth AUV and ROV surveys. It then logically follows that the site descriptions and gas bubble samples obtained from those video surveys that makeup the rest of the Results section were direct results of this decision-making process.

Reply: We incorporated the former section 4.1 into the Material and Method section.

Referee #2: 6. Figure 1 is very helpful in displaying different features of the southern Gulf of Mexico, but the gray and green dots meant to represent probable and definite seeps respectively are hard to distinguish. Although this color scheme is easier to differentiate when the area is magnified in Figure 2, the sites would be better served with different color choices.

Reply: We changed the color of the dots in both Figures 1 and 2 to become better visible.

Referee #2: 7. Figure 3 is clear, but ultimately doesn't contribute much to the manuscript. The text description of identifying gas bubble plumes from multibeam echosounder seems sufficient to communicate the methods of the study to the reader and explain that plumes were not always traceable to the seafloor.

Reply: We removed the Figure.

Referee #2: 8. The dark blue box in Figure 4A showing the ROV survey area is difficult to distinguish from the background bathymetry.

Reply: We changed the color of the box to become more visible.

Referee #2: Typographical errors: - Last sentence of first paragraph of Introduction: "bolder" should be corrected to "boulder."

Reply: Done.

Referee #2: - Same issue in second paragraph of section 4.2.1 ("bolder" instead of "boulder")

Reply: Done.

Referee #2: - Last paragraph of section 4.2.2 (bottom of page 8): "loose buoyancy" should be corrected to "lose buoyancy."

Reply: Done.

Referee #2: - Last paragraph of section 5.1: "temporarily and spatially segregated" should be corrected to "temporally and spatially segregated."

Reply: Done.

Referee #2: - Last paragraph of section 5.3: I believe the authors meant "slow growth" rather than "low growth."

Reply: Done.

Referee #2: - First paragraph of section 5.4: Mictlan Knoll is misspelled as "Mictan Knoll" in the first sentence, and in the third sentence API gravity should be "slightly higher" rather than "slighter higher."

Reply: Done.

---

## Author Comment (AC9) · 29 Jun 2016

Point-by-point reply to Referee #3

Referee #3: I have the following minor comments that would marginally improve the manuscript: 4.2.4 – 'pogonophoran' has fallen out of favor and I am unfamiliar with Anobturata and it is not in WORMs. Can the authors either provide a reference for Anobturata or use frenulate or monolifera (depending on which it may be – with frenulate the more likely of the two and currently more 'correct' (i.e. monophyletic) compared to pogonophoran.)

Reply: We agree and changed it to frenulate.

Referee #3: P13 – "we sketched" suggest "we illustrate"

Reply: Done.

Referee #3: P14 - "As well as oil derived from higher hydrocarbons"– I would suggest changing this to "and likely augmented by degeneration of organic carbon that may include higher hydrocarbons". As Joye et al. found a mismatch between SRR and AOM. They did not completely contribute the sulfide present to longer chain hydrocarbons.

Reply: Done.

Referee #3: P14 "With estimated life spans" – additional age info is found in Cordes et al. 2007 Marine Ecology 28:160-168 that puts smaller individuals (1.1m) at age estimates of 300 years. This provides further evidence for the long term seepage estimates provided by the authors.

Reply: We thank Referee #3 for the comment and included the reference by Cordes et al. (2007). Contrary to Referee #3, Reviewer #1 and #2 suggested to be more cautious in the age estimates based on vestimentiferan tube length. As we cannot contribute to the controversial opinions of using the tube length as chronometers, we follow the suggestion by Referee #1 and #2 and rephrased the paragraph accordingly.